

# On the effect of the uncertainty in soil properties on the simulated hydrological state and fluxes at different spatio-temporal scales

Gabriele Baroni[1,2], Matthias Zink[1], Rohini Kumar[1], Luis Samaniego[1], Sabine Attinger[1,2]

[1] Helmholtz Centre for Environmental Research - UFZ, Department Computational Hydrosystems, Permoserstrasse 15, 04318 Leipzig, Germany
[2] Institute of Earth and Environmental Sciences, University of Potsdam, Karl-Liebknecht-Str. 24-25, 14476 Potsdam, Germany

*Correspondence to*: Gabriele Baroni (gabriele.baroni@ufz.de)

**Abstract.** Soil properties show high heterogeneity at different spatial scales and their correct characterization remains a crucial challenge over large areas. The aim of the study is to quantify the impact of different types of uncertainties that arise from the unresolved soil spatial variability on simulated hydrological states and fluxes. Three perturbation methods are presented for the characterization of the uncertainties in soil properties. The methods are applied at the soil map of the upper Neckar catchment (Germany), as example. The uncertainties are propagated based on the distributed hydrological model mHM to assess the impact of the simulated state and fluxes. The model outputs are analysed by aggregating the results at different spatial and temporal scales. These results show that the impact of the different uncertainties introduced in the original soil map is equivalent when the simulated model outputs are analysed at the model grid resolution (i.e., 500 m). However, several differences are identified by aggregating state and fluxes at different spatial scales (by subcatchments of different sizes or coarsening the grid resolution). Streamflow is only sensitive to the perturbation of long spatial structures while distributed state and fluxes (e.g., soil moisture and groundwater recharge) are only sensitive to the local noise introduced to the original soil properties. A clear identification of the temporal and spatial scale for which finer resolution soil information is (or not) relevant is unlikely to be universal. However, the comparison of the impacts on the different hydrological components can be used to prioritize the model improvements in specific applications, either by collecting new measurements or by calibration and data assimilation approaches. In conclusion, the study underlines the importance of a correct characterization of the uncertainty in soil properties. With that, soil map with additional information regarding the unresolved soil spatial variability would provide a strong support to hydrological modelling applications.

## 1 Introduction

The prediction of mathematical environmental models is affected by uncertainty which arises from inadequate conceptual and mathematical representations of the processes (uncertainty in model structure), inadequate and insufficient knowledge and characterization of system forcing (uncertainty in boundary conditions) and limitations in the measurements or identification of model parameters (parameter uncertainty) (Beven, 2001, 2007; Refsgaard et al., 2007; Tartakovsky et al.,



2012). The need to quantify the predictive uncertainty has led to the development of probabilistic (stochastic) frameworks in many disciplines of environmental sciences and engineering (Altarejos-García et al., 2012; Di Baldassarre et al., 2010; Dubois and Guyonnet, 2011; Savage et al., 2016; Seiller and Anctil, 2014). Nowadays rigorous quantification of uncertainty is an integral part of science-based predictions and decision support systems (Beven, 2007; Farmer and Vogel, 2016; Liu and

Gupta, 2007; Montanari and Koutsoyiannis, 2012).

In hydrological studies, several sources of uncertainty have been studied ranging from atmospheric forcing (Aguilar et al., 2010; Raleigh et al., 2015; Samain and Pauwels, 2013; Vázquez and Feyen, 2003; Zhu et al., 2013) to geology structures (Comunian et al., 2015; Hansen et al., 2014; He et al., 2015; Zech et al., 2015). Among these, the uncertainty related to the soil properties has been widely analysed. Soil properties show in fact high heterogeneity at different spatial scales with a

hierarchy of spatial structures (Burrough, 1983; Heuvelink and Webster, 2001; Vogel and Roth, 2003) and complex interactions with environmental conditions (Lin, 2010). Despite international initiatives exist to improve the current status of soil characterization (Chaney et al., 2016; Heuvelink et al., 2016; Pelletier et al., 2016; Shangguan et al., 2014), detailed information of the spatial heterogeneity of the soil properties over large areas remains a crucial challenge. For this reason, an increasing number of hydrological modelling studies aim to integrate the uncertainty in soil properties that arise from the

unresolved spatial heterogeneity for a proper quantification of the uncertainty of the model results. Since soil properties play a crucial role in the entire water cycle, this topic crosses research fields from lower atmosphere (De Lannoy et al., 2014; Garrigues et al., 2015; Guillod et al., 2013; Osborne et al., 2004; Yu et al., 2014) and surface water (Anderson et al., 2006; Geza and McCray, 2008; Li et al., 2013; Livneh et al., 2015; Salazar et al., 2008) to water and solute transport to groundwater systems (Besson et al., 2011; Hennings, 2002; Yu et al., 2014).

Despite its relevance, however, relative simple assumptions are adopted to characterize the uncertainty in soil properties and to understand its effect on the hydrological response. In several studies the uncertainty is characterized based on relatively small number of scenarios (Baroni et al., 2010; Christiaens and Feyen, 2001; Guber et al., 2009; Herbst et al., 2006; Hohenbrink and Lischeid, 2015; Islam et al., 2006; Mirus, 2015; Moeys et al., 2012) or by a simple random noise (i.e., variance) added to the original soil properties (Arnone et al., 2016; Chaney et al., 2015; Deng et al., 2009; Garrigues et al.,

2015; Han et al., 2014; Loosvelt et al., 2011). Other studies explicitly integrate the complex heterogeneity of the subsurface and the uncertainty in the soil properties is characterized based on spatial correlated random fields i.e., specifying variance and correlation length (Binley et al., 1989; Fan et al., 2016; Fiori and Russo, 2007; Merz and Plate, 1997; Meyerhoff and Maxwell, 2011). Moreover, many of above-mentioned studies focused on the effect of the uncertainty in soil properties on a selected hydrologic variable at specific temporal and spatial scale e.g., rainfall-runoff events (e.g., Arnone et al., 2016; Fan et

al., 2016), simulated evapotranspiration (e.g., Garrigues et al., 2015), soil moisture distributions (e.g., Liao et al., 2014) or groundwater recharge (e.g., Moeys et al., 2012). Simultaneous assessments of different hydrological components of the water balance at different spatial and temporal scales are rare. In addition, due to the different settings used in the studies, it is not possible to draw general conclusions about the role of the uncertainty in soil properties. In some cases the refined spatial information of soil properties does not contribute to a more accurate prediction (e.g., Li et al., 2013). In other studies



the results showed to be very sensitive to the soil properties (e.g., Livneh et al., 2015). These controversial results foster the debate on the need (or not) of finer resolution map in the different modelling applications (Baveye, 2002; Baveye and Laba, 2015; Heuvelink and Webster, 2001).

In the present study, we investigate the impact of the uncertainty of the soil properties on hydrological states and fluxes. The uncertainty in soil properties is characterized by different methods that are consistent in the added noise (i.e., variance) but they differ in the perturbation of the soil spatial structure i.e., correlation length. We hypothesize that local responses of a hydrological system, like evapotranspiration and soil moisture, will be strongly impacted by the uncertainty introduced at small spatial scale. However, integrated responses like the streamflow aggregate local responses over large areas. We hypothesize that this integrated response will be less impacted by soil properties uncertainty. The extent of the impact should decrease with increasing the aggregation area to disappear at a specific domain size. In such a condition, the system is stated to be spatially ergodic as the model output is not any more sensitive to the perturbation i.e., we have the equivalence between spatial and ensemble statistics (Dagan, 1989; Rubin, 2003).

The paper is structured as follow. First, the perturbation methods used for the characterization of the uncertainty of the soil properties are presented. The specific case study is described presenting the catchment, the data used and the specific settings of the perturbation methods. The hydrological model is then introduced together with the uncertainty analysis conducted for the assessment of the effect of the uncertainty in soil properties on the simulated state and fluxes. The results are discussed in section 3 focusing on the effect of the differences detected at different spatial and temporal scales. Final remarks are presented in the conclusions section.

## 2 Methods

### 2.1 Soil perturbation methods

In this section, the three statistical methods to characterize the uncertainty in soil properties are presented. A sketch for describing the methods is provided in Figure 1, where one transect with three soil units characterized by different percentages of sand is shown as example.

The first method (hereafter denoted as Random Error method - RE) is based on the assumption that the nominal value in each soil unit is the only source of uncertainty while the spatial patterns (i.e., soil units) are considered to be correct. To fulfil this assumption, a simple Gaussian random noise is defined with zero mean and given variance (Figure 1, step R1). Random values are sampled from the distribution and added to the nominal value of soil properties of each soil unit (Figure 1, step R2). This approach was commonly used in several studies with the focus of understanding the effect of the soil properties in forward uncertainty analysis of model response (e.g., Deng et al., 2009) or for creating the forward ensemble in data assimilation tests (e.g., Han et al., 2014).

In the second method (hereafter denoted as Spatially Correlated method - SC), a similar assumption of additive random values is considered. However, it is also assumed that the uncertainty arises from the presence of smaller soil units that could





have been not detected in the original soil map (Hennings, 2002). To fulfil this assumption, a spatial structure (i.e., variance and correlation length - CL) is defined (Figure 1, step S1). Based on that, a spatially correlated random field with zero mean is created (Figure 1, step S2) and added to the original soil map (Figure 1, step S3). Random fields are used in this approach to create variability as discussed by Goovaerts (2001) with which simulated short-range components well represent the

complexity of the small-scale spatial structure. Readers interested in the details of the generation of random fields are referred to Deutsch and Journel (1998), Goovaerts (1997) and Isaaks and Srivastava (1989).

Finally, in the third approach (hereafter denoted as Conditional Points method - CP), it is assumed that the nominal value of the original soil units represents some point locations within this unit but their positions are unknown. The uncertainty arises from the spatial variability within these point locations that is not resolved in the original soil map. To fulfil this assumption,

points are randomly distributed over the soil map and the soil properties are associated to each position (Figure 1, step S1). These values are used to calculate the spatial structure i.e., the empirical variogram (Figure 1, step S2). A variogram model is fitted and a conditional random field is created using the sampled locations as conditional points (Figure 1, step S3). It has to be noted that the CP method has some similarity with the pilot points approach used for the calibration of hydrogeological models (Carrera et al., 2005). The main difference is the use in this method of new points at each iteration i.e., the points are

located in different positions for each created conditional random field.

It is noteworthy that additional statistical methods for the analysis of soil map are presented in literature (Goovaerts, 2011; Heuvelink et al., 2016; Kempen et al., 2009; Minasny and McBratney, 2016; Odgers et al., 2014). However, the aim of these methods is to downscale/disaggregate the information available in the original soil map and not to characterize its uncertainty. For this reason, these statistical methods are based on environmental covariates (i.e., environmental variables that co-vary with soil variability) known at higher resolution (i.e., digital elevation model or land use) and they require

relative well knowledge of the soil formation and the specific settings to adopt (Kerry et al., 2012; Nauman and Thompson, 2014; Subburayalu et al., 2014; Du et al., 2015). On the contrary, the three methods selected and developed in the present study represent relative simple approaches only based on the information available in the original soil map. They can be applied for the characterization of any type of soil properties (e.g., texture, saturated hydraulic conductivity, soil depth etc.)

and they reflect different assumptions regarding the uncertainties in the soil properties. For this reason, they can be tuned to characterize uncertainty for soil maps of any scales and they can be easily used to any modelling studies (e.g., sensitivity analysis or data assimilation). Combinations of the methods can be also considered when needed i.e., soil map affected by different types of uncertainties.

## 2.2 Study area

The numerical experiments are conducted in the upper Neckar catchment (Figure 2) that was extensively investigated in previous hydrological studies (Kumar et al., 2010; Samaniego et al., 2010a, 2010b; Wöhling et al., 2013b). This catchment is located in the central uplands of Germany and comprises a catchment area of approximately 4000 km$^2$. The catchment has a gradient in elevation from 250 m to 1015 m a.s.l. with a mean elevation of 550 m. The catchment is prevalently



characterized by cropped field and forest but with a remarkable high degree of urbanization (11%). The long-term mean annual precipitation is around 920 mm a$^{-1}$.

Observed meteorological data, i.e., precipitation as well as minimum, maximum and average daily temperature, were provided by the German Meteorological Service (DWD; www.dwd.de/). These observations have been interpolated to a

4 x 4 km forcing dataset for the hydrological model using external drift kriging. The potential evapotranspiration is estimated using the Hargreaves-Samani method (Hargreaves and Samani, 1985). Data characterizing the land surface are a digital elevation model (Federal Agency for Cartography and Geodesy), a soil map at the scale 1:1000000 (Federal Institute for Geosciences and Natural Resources - BGR), a hydrogeological map (Federal Institute for Geosciences and Natural Resources - BGR), and land cover information (CORINE, European Environmental Agency - EEA, 2009). The soil map

used in the present study (BGR 1:1000000) contains soil texture (percentage of sand, clay and silt) and bulk density [g cm$^{-3}$] for each soil unit. For this study, the vertical soil horizons are aggregated to the total soil depth of 2 m (Figure 3). The map reveals a soil prevalently clay loam but with a relatively high spatial variability represented by 29 soil units of different size within the catchment. All these data are discretized to a spatial resolution of 100 x 100 m$^2$. Readers interested in more details on data-set and the processing may refer to Kumar et al. (2010), Samaniego et al. (2010b) and Zink et al. (2016). The spatial

distributions of cumulative rain, potential evapotranspiration, land use and the mean annual leaf area index are shown in the supplementary material (see Figure S1).

## 2.3 Settings of the soil perturbation methods

In this section, the specific settings of each statistical perturbation method used for the characterization of the soil properties are described. The three methods are used independently to generate three different ensembles to identify the impact of the

different uncertainties introduced in the original soil map on simulated state and fluxes.

Considering the random error method (see Figure 1), a Gaussian random additive noise is used with variance 50 [%$^2$] and 0.05 [g$^2$ cm$^{-6}$] for soil texture (sand and clay) and bulk density, respectively (Table 1). A correlated sampling design is used to preserve the correlation between the original soil properties (e.g., negative correlation between sand and clay) and the values are forced to a realistic range i.e., not negative texture values and sum of textural fractions does not exceed 100%.

These variances are selected to perturb the soil properties within the original soil class i.e., it is assumed that the exact values of the soil properties are unknown but the soil class (e.g., clay loam) is correct. Similar ranges were also applied in other studies (Han et al., 2014; Hennings, 2002).

For the spatially correlated method (see Figure 1), the parameters for the variogram and co-variogram models are selected to be consistent with the perturbation introduced in the random error method (Table 1). In particular, exponential variogram

models are prescribed with the same effective variances used in the random error method (i.e., 50 [%$^2$] and 0.05 [g$^2$ cm$^{-6}$] for texture and bulk density, respectively) and preserving the correlation between the original soil properties. The correlation length of 3 km is selected to represent relative small spatial patterns that were not captured by the original soil map i.e., area



smaller than most of the soil units (Figure 3). The variogram and co-variograms models selected are shown as supplementary material (see Figure S2).

Finally, considering the conditional points method (see Figure 1), tests are conducted to identify the density of the conditional points within the soil map. One sample at every 3 x 3 km$^2$ is found to introduce the same variance prescribed by

the other two methods (Table 1). A stratified spatial random sample is used to distribute the points within each soil units. Based on these, two nested exponential variogram and co-variogram models are fitted to the experimental variograms based on ordinary least-squares residuals (Pebesma, 2004). These variogram models are used to create the conditional random fields. The experimental variograms and the fitted models for one realization (i.e., one random field) are shown, exemplarily, in the supplementary material (Figure S3).

For each method, an ensemble of 100 realizations is created to characterize the uncertainty in soil properties. The analysis is conducted with the statistical software R 3.2.x (R Core Team, 2013) and its packages (Pebesma, 2004). The multi-variate conditional random fields were generated with GCOSIM3D code (Gómez-Hernández and Journel, 1993).

## 2.4 The hydrological model mHM

The effect of the uncertainty in soil properties as characterized by the three perturbation methods on hydrological states and

fluxes is analysed using the hydrological model mHM. The mesoscale Hydrological Model mHM (Kumar et al., 2013; Samaniego et al., 2010b) is an open source, spatially distributed hydrologic model (www.ufz.de/mhm). It considers interception, snow accumulation and melting, soil water retention, evapotranspiration, percolation, and runoff generation as main hydrologic processes. The Multiscale Parameter Regionalization (MPR) embedded in mHM allows for the application of the model at various locations and scales (Kumar et al., 2013; Rakovec et al., 2016). MPR accounts for sub-grid

variabilities by estimating model parameters at the scale of the morphological input, e.g. 100 x 100 m$^2$. Subsequently, these parameters are upscaled to the model resolution. For a detailed model description and the regionalization scheme interested readers may refer to Samaniego et al. (2010b) and Kumar et al. (2013). For this study, the soil within mHM is discretized into 3 layers ending in 5 cm, 25 cm, and a variable depth below ground. The depth of latter is based on the information provided by the soil map (2 m). Based on the soil textural properties, mHM estimates effective parameters for porosity,

hydraulic conductivity, field capacity and permanent wilting point using a set of pedotransfer functions (e.g., Zacharias and Wessolek, 2007).

The model was calibrated and validated in previous studies showing very good capability to match streamflow measurements at catchment of different sizes (Kumar et al., 2010, 2013; Samaniego et al., 2010b; Wöhling et al., 2013b). The same parameterization is used for the present study. We establish the mHM over the Upper Neckar catchment at 500 m

spatial resolution, which covers around 16430 grid cells. The model run is conducted at an hourly time scale. All simulations are conducted with a five year model spin up time (1985 - 1989) to minimize the effect of inappropriate initial conditions. The implications of uncertain soil properties are evaluated showing the uncertainty in simulated routed streamflow (*SF*), generated runoff at every grid cell (*Q*), actual evapotranspiration (*AET*), volumetric soil moisture (*SM*) in the upper 30 cm





and groundwater recharge (*GWR*). For each perturbation method 100 simulations were performed which yield in a total of 300 simulations. The results obtained during one year of forward simulation (1990) are shown, as example. This year is selected to represent average climate condition of the area (i.e., two rain seasons concentrated in spring and fall and a relatively dry summer season) but with a relatively high variability within the catchment (see Figure S1 in supplementary material).

## 2.5 Uncertainty analysis at different spatio-temporal scales

The uncertainty in simulated states and fluxes is quantified based on the coefficient of variation (*CV* [%]) to allow comparability between the results obtained in the different model outputs. Assuming a generic variable *v* representing simulated state or fluxes, *CV* is calculated as follow:

$$CV_{i,t}^m = \frac{\sigma_{i,t}^m}{\mu_{i,t}^m} \, 100 \tag{1}$$

where $\sigma$ is the standard deviation of the variable *v* at each cell *i* and time *t* calculated based on each perturbation method *m* (i.e., random error method, spatially correlated method or conditional points method) as follow:

$$\sigma_{i,t}^m = \sqrt{\frac{1}{N_{ens}} \sum_{j=1}^{Nens} \left( v_{i,t}^{m,j} - \mu_{i,t}^m \right)^2} \tag{2}$$

with $N_{ens}$ the number of ensemble members (i.e., 100), *j* one single ensemble member and $\mu$ representing the mean of the ensemble at each cell *i* and time *t* calculated as follow:

$$\mu_{i,t}^m = \frac{1}{N_{ens}} \sum_{j=1}^{Nens} v_{i,t}^{m,j} \tag{3}$$

The values obtained with the three perturbation methods are compared by aggregating the simulated states and fluxes at different spatial and temporal resolutions. In particular, four analyses are conducted (Table 2).

In the analysis #1, the spatial variability of the uncertainty of the simulated state and fluxes is presented i.e., depending on the geographical location within the catchment. In this case the average *CV* calculated for the entire simulation period (i.e., one year) in each grid cell is quantified as follow:

$$\overline{CV}_i^m = \frac{1}{T} \sum_{t=1}^{T} CV_{i,t}^m \tag{4}$$





where $T$ is the number of simulations time steps (i.e., 365 days). This value is used to represent and discuss the average uncertainty obtained in the specific cell $i$ and its spatial variability within the catchment.

In the analysis #2 (Table 2), the daily temporal dynamic of the uncertainty obtained at each grid cell is discussed. For this reason the $CV_{i,t}^m$ calculated at the daily time step (Eq. 1) is directly compared for two representative grid cells selected within

the catchment.

The uncertainty on simulated states and fluxes is further compared by aggregating the model outputs at different resolution to identify the effect of the spatial scale on the performance of the model as discussed by Refsgaard et al. (2016). In particular, for the analysis #3 (Table 2), subcatchments of different sizes are defined (see Figure 2) and the effect of the uncertainty in soil properties to the streamflow routed to the outlet (SF) of each subcatchment is compared. For the other

simulated model outputs (i.e., evapotranspiration, soil moisture and groundwater recharge), the values of each grid cell within the subcatchment are aggregated calculating the average of simulated model output $v$ obtained at the finer resolution as follow:

$$\bar{v}_{sc,t}^{m,j} = \frac{1}{N_{sc}} \sum_{i=1}^{N_{sc}} v_{i,t}^{m,j} \tag{5}$$

where $N_{sc}$ is the number of grid cell within the subcatchment $sc$. The value $\bar{v}_{sc,t}^{m,j}$ is used in Eq. (1 - 4) to calculate and compare the coefficient of variation of the mean simulated state and fluxes for the subcatchments of different sizes.

Finally, in the analysis #4 (Table 2), the effect of the aggregation of states and fluxes at different resolutions is further analysed based on the approach showed by Hansen et al. (2014) and Rasmussen et al. (2012). In this case, the generic model

output $v$ is averaged coarsening the model grid at different resolutions $r_d$ (i.e., $r_2 = 2$ km, $r_4 = 4$ km, $r_8 = 8$ km, $r_{16} = 16$ km, $r_{32} = 32$ km). These values are substituted in Eq. (1 - 3) to calculate the coefficient of variation in each new coarsened grid cell $i$. In this analysis the average of the $CV$ across the entire domain and over the entire simulation period (i.e., 365 days) is calculated as a summary statistic as follow:

$$\overline{CV^{m,r_d}} = \frac{1}{T} \sum_{t=1}^{T} \frac{1}{N_r} \sum_{i=1}^{N_r} CV_{i,t}^{m,r_d} \tag{6}$$

with $N_r$ representing the number of cell $i$ within the coarsened domain $r_d$.

In addition to the spatial dimension, in this study, the same procedure is also repeated for each spatial aggregation $r_d$ considering a time aggregation $t_d$. In particular, all the simulated model outputs $v$ obtained at daily time step are averaged at

$t_{10} = 10$ days, $t_{30} = 30$ days, $t_{60} = 60$ days, $t_{120} = 120$ days and $t_{180} = 180$ days, respectively. These values are substituted in Eq. (1-4) to calculate the coefficient of variation in each temporal aggregation $t_d$ and they are considered to represent the uncertainty in the simulated state and fluxes in case the averaged values are used in the assessment of the performance of the model.





The four analyses described above are conducted based on the results of 100 simulations obtained with the distributed hydrological model for each perturbation methods. A total of 300 simulations, analysed in 12 cases, are discussed in the results section (Table 2).

## 3. Results and discussion

### 3.1 Perturbation of the soil properties

Three methods are used to perturb the values of the original soil map i.e., sand [%], clay [%] and bulk density [g cm$^{-3}$]. This section discusses the results obtained on clay percentage exemplarily. Similar results are obtained for the other soil properties (see supplementary material, Figure S4 - S7). For each method, Figure 4 (top row) shows one realization of the perturbed clay percentage. In addition (Figure 4, down row), one transect along the catchment is selected and the clay percentage of the original soil map, of one realization and of the ensemble spread (95% confidence interval) are shown. The longitudinal transect was selected to capture the strong variability in the soil units detected along this direction (see Figure 3).

The random error method (RE) preserves the shapes of the soil units and perturbs just the nominal values. The results therefore show how the contrasts between the soil units are modified and in some cases are exaggerated. For this reason, it is noteworthy to observe that this method could create non-realistic spatial patterns since soil properties usually show smother changes in space. The results obtained based on the spatially correlated method (SC) show that the shapes of the soil units are still highly identifiable and, with that, still the sharp changes between the units are preserved. With this method, however, the random fields superimposed to the original soil map were selected with a correlation length of 3 km (see section 2.3). For this reason, smaller spatial structures than the original soil units are introduced and the sharp changes in the soil properties are not uniformly distributed all over the soil unit. Finally, considering the results obtained with the conditional points method (CP), the results show that the soil units are visible but the contrasts are completely smoothed eliminating the artefact of the original soil map. However, a wider spread in the ensemble in comparison to the perturbation obtained within the unit is visible in this transition between the soil units. The effect is due to the combination of the uncertainty introduced to the nominal value of the soil property and to the exact position of the transition between the soil units.

The spread of the realizations is quantitatively evaluated based on the standard deviation of the ensemble. In particular, Figure 5a represents the probability distribution of the standard deviation of the clay percentage calculated at every grid cell within the catchment (i.e., 16432 grid cells) for each method. Results obtained based on the three methods, on average, exhibit a high consistency in representing the uncertainty over the catchment (i.e., average standard deviation is for all the methods 7%). However, some differences are detected in the distributions. The random error method (RE) shows a normal distribution with a relatively low variability (i.e., the coefficient of variation of the distribution is 6%). This is the consequence of the fact that the soil properties within the catchment are perturbed with almost the same magnitude. Similarly, the spatially correlated method (SC) shows also a normal distribution but with a slightly wider variability (i.e.,



$CV$ = 8%). In contrast, the results obtained with the conditional points (CP) method show a very different distribution that is skewed with a tail to high extreme values. These high spreads in the soil realizations are located in the transition between the soil units, in particular if the transition is sharp (see Figure 4).

Finally, the standard deviation of the values is calculated by aggregating the map for different subcatchments (Figure 5b) and at different grid resolution (Figure 5c) based on the analysis detailed in section 2.5. The spreads of the realizations obtained with the three perturbation methods are of similar magnitude considering the finer resolution (e.g., resolution < 1 x 1 km$^2$). The differences between the perturbation methods become more relevant by aggregating to a coarser resolution. In other words, this means that the introduced uncertainty is in the same order of magnitude but the spatial patterns are different. In the random error method (RE), the spread is relatively high at all scales and even the mean value of the soil properties of the entire catchment is perturbed (i.e., resolution of 60 x 60 km$^2$). The spread obtained with the spatially correlated method (SC) decreases more rapidly with increasing the spatial scale. This is consistent with the correlation length prescribed to the random fields used in this method (i.e., 3 km). However, it is noteworthy to observe how also the average over the entire catchment is still perturbed (i.e., standard deviation > 0 also for the resolution of 60 x 60 km$^2$). This is explained by the fact that the random fields superimposed to the original soil map have zero mean over a rectangular domain but the average can be different when masked to the catchment. The behaviour is exaggerated when a relatively long correlation length in comparison to the size of the domain is used. Finally, the results of the conditional points method (CP) show how just the small scale is perturbed and the spread of the ensemble drops already at the resolution of 5 km to disappear completely when the average over the catchment is considered. This behaviour is consistent with the density of the samples used to constrain the random fields (i.e., one sample every 3 x 3 km$^2$).

## 3.2 Spatial variability of the uncertainty of state and fluxes

In this section, the spatial variability of the uncertainty of the simulated state and fluxes is presented. In this analysis (see section 2.5, Table 2, uncertainty analysis #1), the mean coefficient of variation over time (i.e., 1 year) is calculated for each grid cell (i.e., 16432 grid cells) and the spatial distributions obtained with the three perturbation methods are compared (Figure 6).

The uncertainties of all hydrological states and fluxes obtained with each perturbation methods provide nearly the same magnitude and the same spatial variability, with correlation coefficients calculated between the results obtained by each method higher than 0.8. For this reason, only the spatial distribution of the coefficient of variations ($CVs$) of the model outputs over the entire catchment obtained with the random error method (RE) is shown as example (Figure 6, left). The results obtained with all the three perturbation methods are shown only for the same transect depicted in Figure 4 (Figure 6, right) to facilitate the visualization of the relatively small differences.

In general, the results obtained based on all the three perturbation methods show that the uncertainty in the total runoff ($Q$) and groundwater recharge ($GWR$) are highest, with an average $CV$ estimated over the catchment of 15% and 11%, respectively. Soil moisture ($SM$) and actual evapotranspiration ($AET$) appear to be less sensitive to the soil variability with an



average *CV* of 3% and 1%, respectively (Figure 6, left). The relatively small differences detected based on the use of different perturbation methods are located in the transition between the soil units (Figure 6, right) and they are attributed to the higher uncertainty in the soil properties introduced in those areas (see Figure 5a). Overall, a strong spatial variability in the uncertainty in the model outputs is detected with some differences depending on the considered model output. The
uncertainty in runoff is more pronounced in the north-west areas, actual evapotranspiration appears to be more affected on the central-north areas. High uncertainty in simulated soil moisture is distributed across the catchment and the uncertainty in simulated groundwater recharge increases close to the catchment outlet (see Figure 2).

For a further interpretation, the spatial variability of the uncertainty in the simulated model outputs is compared to different boundary conditions and input properties. In particular, the correlation coefficients between the spatial distribution of the
*CVs* of each model outputs and the spatial distribution of clay [%], the mean leaf area index (LAI [$m^2\,m^{-2}$]) and the annual sum of the potential evapotranspiration (*PET* [mm]) calculated over the simulation period (i.e., one year) are calculated. These three factors are selected to represent soil, vegetation and atmospheric conditions, respectively. The spatial distributions of these factors are shown in the supplementary material (see Figure S1).

The results obtained with the three different methods are consistent between each other also in this comparison (i.e., as
represented by the small error bars) showing different correlations for each model output (Figure 7). The uncertainty in the runoff is stronger correlated to the actual value of the soil property. This correlation can be visually identified comparing the spatial variability detected in Figure 6 (right) and the spatial variability of the soil property shown in Figure 4 for the same transect. The uncertainty in the actual evapotranspiration is strongly correlated to the atmospheric conditions and, to less extend, to the soil properties. Finally, the uncertainties of the soil moisture and groundwater recharge are correlated to the
vegetation characteristics, with a relatively lower effect of soil properties.

To further evaluate the different correlations found for each simulated model output, the correlation matrix between the uncertainty (*CV*) detected in each model output is calculated (Table 3). On the one hand, the results show that the uncertainties in the fluxes are positively correlated (correlation coefficient > 0.2). This means that when the uncertainty in one specific flux is relatively high, also other fluxes to some degree are uncertain. On the other hand, it is interesting to note
that the uncertainty in soil moisture is highly correlated to the groundwater recharge (correlation coefficient = 0.7) while the correlations to the other model outputs are negligible (correlation coefficient < 0.2). This means that the model could have relatively low uncertainty in soil moisture but high uncertainty in evapotranspiration or runoff and vice-versa. These results are consistent among all the three perturbation methods and they support the use of both state and fluxes for a proper assessment of the performance of hydrological models as it was underlined in several other studies (Ahmadi et al., 2014;
Baroni et al., 2010; Conradt et al., 2013; Delsman et al., 2016; McCabe et al., 2005; Rakovec et al., 2016; Silvestro et al., 2015; Wöhling et al., 2013a; Zink et al., 2016).





### 3.3 Temporal variability of the uncertainty of state and fluxes

The daily temporal variability of the uncertainty on the simulated state and fluxes obtained at the model resolution (i.e., 500 m) is presented in this section. In this analysis (see section 2.5, Table 2, analysis #2), the coefficient of variation at daily time step for each perturbation method obtained in two grid cells selected within the catchment are compared for an

illustrative purpose. The two locations A and B are depicted in Figure 2. The two grid cells are characterized by (see also supplementary material, figure S1) a remarkable difference in the rain (i.e., almost 1600 mm a$^{-1}$ and 1000 mm a$^{-1}$, respectively), by different land use (i.e., crop field and deciduous forest, respectively) but they have almost the same soil properties (i.e., 19% sand and 59% clay for grid cell A; 19% sand and 66% clay for grid cell B). The grid cells are selected to represent different uncertainties of the model outputs (see Figure 6). In particular, grid cell A shows relatively high

uncertainty in simulated soil moisture ($CV \sim 4\%$) while grid cell B show relatively low uncertainty ($CV \sim 2\%$). The three perturbation methods provide nearly the same results with a correlation coefficient higher than 0.8. For this reason, only the results obtained with the random error method (RE) are shown in Figure 8. The figure also shows the actual values of simulated state and fluxes for comparison (i.e., mean value and 95% confidence interval of the ensemble simulations obtained with the random error method).

The results show how the uncertainty of the total runoff is relatively high during the entire simulation period with a tendency of increasing the uncertainty during high flow period. The behaviour is particularly evident in the grid cell B (i.e., correlation coefficient between $CV$ and simulated runoff is 0.6). In contrast, the actual evapotranspiration is close to the potential rate for most of the simulation period and, for this reason, it is not sensitive to changes in soil properties. As expected, the uncertainty is only detected during summer time when soil moisture is relatively low and the actual evapotranspiration rate

decreases in comparison to the potential evapotranspiration. The temporal variability obtained for the uncertainty in soil moisture shows a more complex behaviour depending on the grid cell considered. In grid cell A, the $CV$ increases with the increasing of soil moisture while it decreases in grid cell B. The different behaviours are explained comparing the actual soil moisture values. In the first grid cell, the soil moisture values are relatively low (0.25 m$^3$ m$^{-3}$) while, in the second cell, the values are close to saturation (0.4 m$^3$ m$^{-3}$). Finally, groundwater recharge shows also a strong temporal dynamic with a

tendency of higher uncertainty with increasing groundwater recharge in the grid cell A (correlation coefficient = 0.2) while the correlation is negligible in grid cell B (correlation $\sim 0$).

Overall, it is noteworthy to observe how the uncertainty in soil moisture is relatively constant in time while the uncertainty in the fluxes shows much stronger temporal variability. This different behaviour can be explained considering two main characteristics. On the one hand, the presence of non-linear relations between state and fluxes generates threshold behaviour

for which the uncertainty in soil moisture could be limited to ranges where the fluxes are not affected. This is for instance the case with high soil moisture and evapotranspiration or low soil moisture and runoff generation. On the other hand, there is a tendency of compensation in the uncertainty in the model outputs for which an overestimation of the actual evapotranspiration could be related to an underestimation of the groundwater recharge (or vice-versa). In these conditions



the soil moisture could be still well defined without providing any indication of the degradation of the model performance. As a result, the low uncertainty in soil moisture does not represent the overall uncertainty in the model. Overall, this analysis underlines the role of the different hydrological conditions (e.g., dry or wet) for understanding the effect of the uncertainty in soil properties on the model response. Similar conclusions are supported by the use of temporal sensitivity and identifiability

analysis to better capture the role of the different uncertainties in the parameters analysed (Ghasemizade et al., 2017; Guse et al., 2015; Pianosi and Wagener, 2016; Wagener et al., 2003).

### 3.4 Spatial uncertainty of state and fluxes at subcatchments

The uncertainties (*CV*) of simulated state and fluxes are also compared by aggregating the results over subcatchments of different sizes (see section 2.5, Table 2, analysis #3). The results obtained with the three perturbation methods are shown

against the catchment size in Figure 9.

As presented by Refsgaard et al. (2016), the uncertainty in all the model output reduces with increasing catchment area. However, it is interesting to note that the three perturbation methods generated very different results. The random error method (RE) creates higher uncertainty in all the subcatchments and even the mean of states and fluxes over the entire catchment is uncertain (i.e., 60 x 60 km$^2$). The spatial correlated method (SC) shows a similar pattern but the uncertainty is

lower in all the subcatchments. Finally, the uncertainty based on the condition points method (CP) decreases already at small catchment sizes of e.g., 2 x 2 km$^2$.

The different results obtained with the three perturbation methods have important implication when considering the specific model application. For instance, it is notable how the streamflow at the catchment outlet, that was used for calibration of the model in previous studies (Kumar et al., 2013; Samaniego et al., 2010b), is sensitive only to the perturbation of long soil

spatial structures introduced with the random error method. In contrast, the streamflow at the catchment outlet is not sensitive to the perturbations introduced at small scale (e.g., conditional points method). On the one hand, this means that small soil variabilities are not relevant when the model application focuses on the streamflow prediction. On the other hand, this results underlines that it is not possible to infer (e.g., calibrate) these small spatial soil patterns based on the streamflow observations. The contrary behaviour is noted for the distributed hydrological states and fluxes (evapotranspiration and

groundwater recharge). These distributed model output represent in fact local conditions. For this reason, they show to be sensitive to all type of perturbation introduced. This means that these localized state and fluxes can be used to infer local properties but it is not possible to use this type of observations to calibrate the values for larger areas. For this reason, the use of e.g., remote sensing products as total water storage anomalies and evapotranspiration is an effective approach for constraining and improving model parameterization (e.g, Rakovec et al., 2016).

The different results in the uncertainty in the model outputs obtained by the use of the different perturbation methods are consistent with the different uncertainties introduced in the soil properties (Figure 5b). This result supports the conclusion that the differences are related to the underlying correlation length scale used in each perturbation method (Refsgaard et al., 2016). The random error method perturbs the value of the entire soil units and it does not generate spatially ergodic soil



parameters fields i.e., aggregated hydrological responses still show a non-vanishing uncertainty at large catchment. The spatial correlated method introduced correlation length of 3 km and the effect on the uncertainty in the aggregated model output reached a remarkable reduction (e.g., > 90% of the uncertainty in all the simulated state and fluxes is reduced) when the entire catchment is considered. Finally, the conditional points method introduces uncertainty only at small spatial scales

while the longer spatial patterns are preserved. For this reason the domain is ergodic already at relatively low catchment size (i.e., 20 x 20 km$^2$).

These characteristic lengths (catchment dimension D *vs.* correlation length CL) identified by the use of the three different soil perturbation methods are in agreement with previous studies conducted in surface hydrology (Binley et al., 1989; Fan et al., 2016; Herbst et al., 2006; Merz and Plate, 1997) and in stochastic subsurface hydrology (Dagan, 1989; Fiori and Russo,

2007; Rubin, 2003), where a suitable value for defining ergodic system (or representative scale) was found to be ~ D/CL > 20. However, two important characteristics can be further underlined. First, it is notable a certain spread in the uncertainty of catchment with similar size. This behaviour is in agreement with the results discussed in section 3.2 showing different sensitivity on the soil perturbation depending on the different boundary conditions and model set-up (i.e., depending on the location within the catchment). For this reason, the results support the difficulties to find a universal

representative scale that is not affected by the uncertainty in the soil properties for the entire catchment. Despite this scale has some differences with the Representative Elementary Area concept (REA) introduced in past literature (see Refsgaard et al. (2016) for further discussion about the differences), it is noteworthy how this result is in agreement with the difficulties for finding a universal REA discussed also in those studies (e.g., Fan and Bras, 1995; Wood et al., 1988). Secondly, different sensitivities arise depending on the model output considered. Soil moisture is more sensitive to the perturbation of soil

properties since the relative change between the three different methods is highest among the four hydrological variables under investigation. This behaviour is particularly evident when considering the results obtained with the random error method. In this case, a relatively small perturbation introduced in the mean of the entire catchment (60 x 60 km$^2$) explains already most of the uncertainty in the simulated soil moisture. The uncertainty slightly increases with reducing the catchment size. In comparison, all the fluxes are much less affected by the small perturbations introduced for the entire catchment but

they become increasing pronounced with a decreasing catchment size. For this reason, the representative scale is also different depending on the model output considered.

### 3.5 Uncertainty of state and fluxes at different spatio-temporal scales

A similar scaling analysis is also conducted averaging state and fluxes by coarsening the grid resolution and by aggregating at different temporal scales (see section 2.5, Table 2, analysis #4). The results obtained with the three perturbation methods

are presented in Figure 10.

The spatial aggregation of the model output, as represented in the x-axis of Figure 10, shows the same effect obtained by aggregating the model output based on catchment of different sizes (Figure 9). For this reason, the two analyses (aggregating by catchment *vs.* coarsening the grid resolution) can be considered equivalent in the identification of the effect of the spatial



resolution on the uncertainty in the model outputs. However, the results described in the previous sections showed also a strong variability in space and in time. For this reason, the use of the mean coefficient of variation calculated over time and across the all number of grid cells to represent the model performance (Eq. 6) can be misleading e.g., underestimating the actual uncertainty in the model output. Instead, the use of the maximum $CV$ calculated over the catchment and over the

simulated period could be used to better represent the model performance. In addition, the extension of the analysis to the temporal scale (y-axis in Figure 10) emphasizes the clear trade-off of the performance of the model between the spatial resolution and the temporal resolution. In particular, assuming an arbitrary threshold as a limit of model predictive capability for the specific model application (Refsgaard et al., 2016), the spatial and temporal analysis shows how the simulated state and fluxes should be aggregated in time to maintain acceptable performance when decreasing the space resolution. On the

one hand, this analysis can support the use of the model results for specific model applications. On the other hand, it would be possible to identify when it might become important to have a better representation of the soil spatial variability for the improvement of the performance of the model.

Finally, it is noteworthy how these conclusions are supported by the results obtained with all the three perturbation methods. However, the actual scale at which it might become important (or not) to have a better understanding of the soil spatial

variability strongly depends on the perturbation methods used. Since the three perturbation methods reflect different uncertainties introduced in the original soil map, the analysis enphasises the importance to identify the correct approach for each model application. For the specific case study presented here, it could be assumed that the uncertainty in soil properties that affects the simulated streamflow at the catchment outlet is well compensate by the calibration (Kumar et al., 2013; Samaniego et al., 2010b). For this reason, the random error method could not represent the actual uncertainty in the specific

model application as it shows to strongly effect also the simulated streamflow. The same could be considered for the results obtained with the spatially correlated method, as soon as subcatchment of different sizes are used in the calibration. On the contrary, the conditional points method appear to be a simple and effective method to preserve the general spatial pattern of the original soil map while introducing uncertainty due to the unresolved spatial heterogenity within the soil units. This type of uncertainty affects the streamflow only for small subcatchments (size < 1 x1 km$^2$) while introducing relevant effects on

the local hydrological states (i.e., soil moisture) and fluxes (e.g., grondwater recharge). This method therefore could be considered as a valuable choice to account for the uncertainty of soil properties for this type of model applications i.e., when well calibrated hydrological models based on streamflow measurements are used.

## 4. Conclusions

In the present study, the uncertainty in soil properties is characterized based on three statistical perturbations methods. This

uncertainty is propagated applying the distributed hydrological model mHM. The uncertainty in the simulated states and fluxes are analysed at different spatial and temporal scales. The main conclusions are summarized as follow.



1.  The effect of the uncertainty in soil properties depends on the hydrological model output. In particular, the uncertainty in the fluxes are relatively positive correlated i.e., if the uncertainty in one of the simulated flux is high, also the other fluxes show, to some degrees, uncertainties. On the contrary, the uncertainty in the simulated soil moisture shows a more complex relation as its uncertainty does not always represent the overall uncertainty in the simulated fluxes. This behaviour is explained by the non-linear relation between state and fluxes and the occurrence of threshold conditions in the model response. For this reason, these results support the need of multi-variable (e.g., soil moisture and streamflow) for a proper assessment of the overall performance of hydrological models (Rakovec et al., 2016; Zink et al., 2016) and the use of temporal diagnostic tools for a better understanding of the input-output space (Ghasemizade et al., 2017; Guse et al., 2015; Pianosi and Wagener, 2016; Wagener et al., 2003).

2.  The uncertainty in state and fluxes depends on the specific locations and on the boundary conditions. In particular, the uncertainty in the model results shows strong temporal and spatial variability over the catchment with complex interactions to local environmental conditions (i.e., atmosphere, vegetation and soil). These results highlight the role of specific model settings (i.e., parameters and boundary conditions) for a proper characterization of the model response and the difficulty to generalize the result for other applications (i.e., different study areas). Similar conclusions were obtained based on sensitivity analysis conducted using hydrological models in different catchments (e.g., Shin et al., 2013; van Griensven et al., 2006).

3.  The uncertainty in state and fluxes depends on the spatio-temporal resolution used for the analysis. In particular the uncertainty in all the model outputs decreases with increasing the spatial and temporal resolution. Assuming an arbitrary threshold acceptable for a specific model application, this scaling analysis identifies, on the one hand, the spatial and temporal resolution at which the model output could be used. This resolution is referred as Representative Elementary Scale (RES) by Refsgaard et al. (2016) and it provides a clear and simple framework for the assessment of the performance of distributed models. On the other hand, this Representative Elementary Scale identifies the resolution below which it might become important to have a better understanding of the soil spatial variability. Two possible extensions of the RES approach as proposed by Refsgaard et al. (2016) are the use of the maximum $CV$ and the temporal aggregation. The former should better capture the model performance due to the strong spatial and temporal variability that could be present in the uncertainty within the catchment. The latter could be used to emphasize the trade-off between temporal and spatial resolution of the model application.

4.  The assumptions and the methods used for the characterizations of the uncertainty in the soil properties plays a crucial role. In particular, the above conclusions are supported by the results obtained with all the three soil perturbation methods used in this study. However, the absolute value of the uncertainty detected in state and fluxes at different spatial and temporal scales strongly depends on the perturbation methods. For this reason, the results underline the importance to properly characterize the specific sources of uncertainty to transform a pure numerical exercise to results with physical sound that are able to better support the model applications. The three methods developed and used in the present study represent three simple approaches that can be considered to account for



different types of uncertainty in the soil map. In this context, however, the availability of soil map with additional information regarding not only the actual mean value within the soil units but also information representing the unresolved variability (variance and correlation length of the subdominant soil units) would provide a strong support to hydrological modelling applications.

5.  Finally, the analysis conducted in the present study identifies important information to be used for possible model improvement, either by collecting additional data regarding the soil properties or for inverse modelling and data assimilation frameworks. In particular, integrated fluxes like river discharge of large catchments show to be not impacted by small scales soil variabilities (i.e., variance) but only by long spatial structures (i.e., long correlation lengths). For this reason, additional details in the soil map do not improve the model performance but rather other sources of uncertainties should be considered for that (e.g., vegetation properties). For the same reason, this integrated observation cannot be used to infer local parameters (i.e., parameter of finer resolutions) but only mean characteristics of the input factor (e.g., average soil properties over the soil units). On the contrary, local state and fluxes show to be very sensitive to local variation in the soil properties (i.e., variance). For this reason, soil map with finer resolution data is found to be an important factor for further improvement of the performance of the model. For the same reason, these simulated outputs can be used to infer local soil parameters in calibration or data assimilation. Despite the transition between these two extreme conditions for which the uncertainty in soil properties is (or not) important is quite smooth, it depends on the output considered and on the boundary conditions, this analysis provides a strong support to prioritize the model improvements in specific model applications. For this reason, similar studies can be considered for comparing statistical methods to characterize other sources of uncertainty relevant in catchment hydrology (e.g., precipitation, vegetation parameters).

*Acknowledgment.* The study was supported by the Deutsche Forschungsgemeinschaft (DFG) under CI 26/13-1 in the framework of the research unit FOR 2131 "Data Assimilation for Improved Characterization of Fluxes across Compartmental Interfaces" and by the Helmholtz Alliance - Remote Sensing and Earth System Dynamics (HGF-EDA).

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


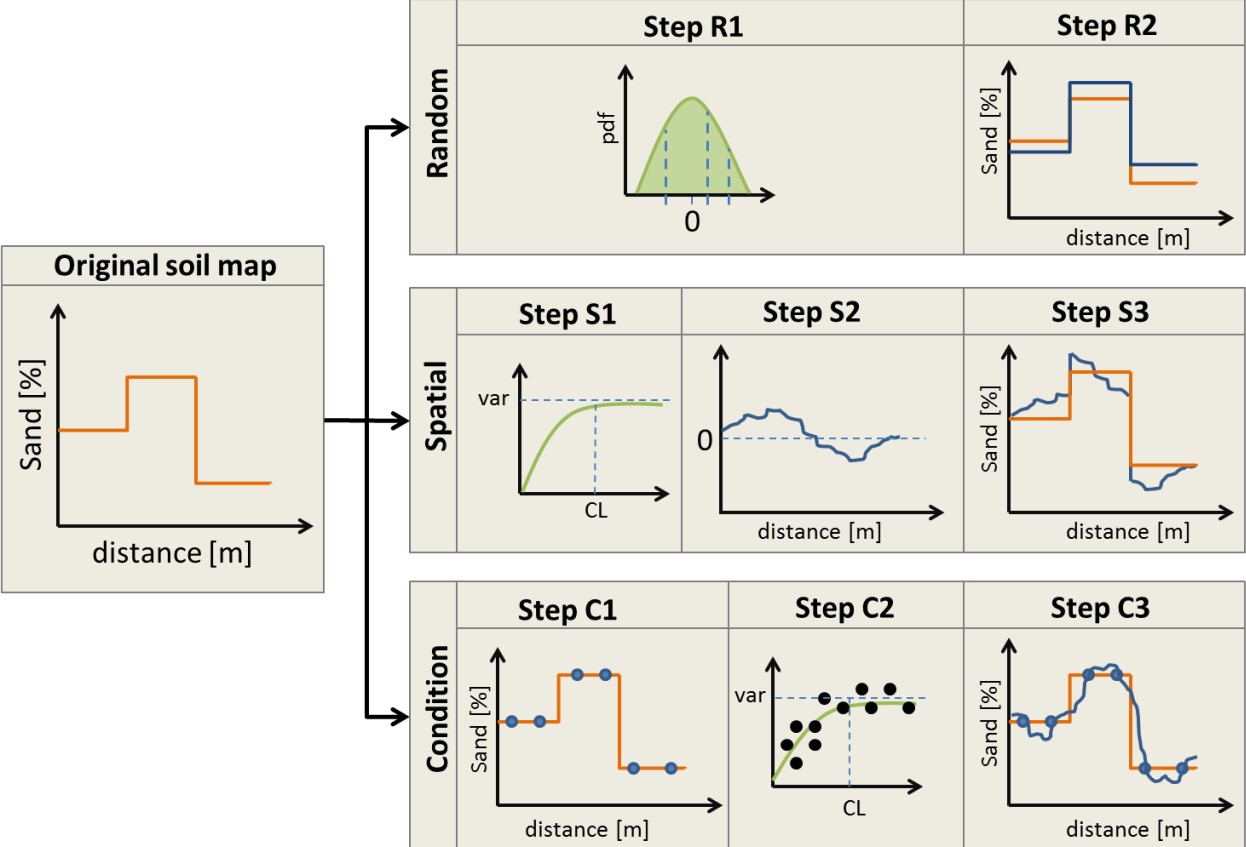

**Figure 1.** Soil perturbation methods (Random, Spatial and Condition). The panel on the left shows the fraction of sand of the original soil map as orange line. Within the transect three different soil units are observed, which leads to three different sand contents. Each row of the right panels depicts the steps for setting the perturbation methods. The blue line depicts one realization of the respective perturbation method. The detailed description of these methods can be found in section 2.1. Abbreviations: var – variance, CL – correlation length.





**Figure 2.** Location of the upper Neckar catchment within Germany. The positions of the 36 gauging stations (red points) used for defining the subcatchments, the transect (dashed black line) and the two grid cells analysed (green points A and B) are depicted on the map.





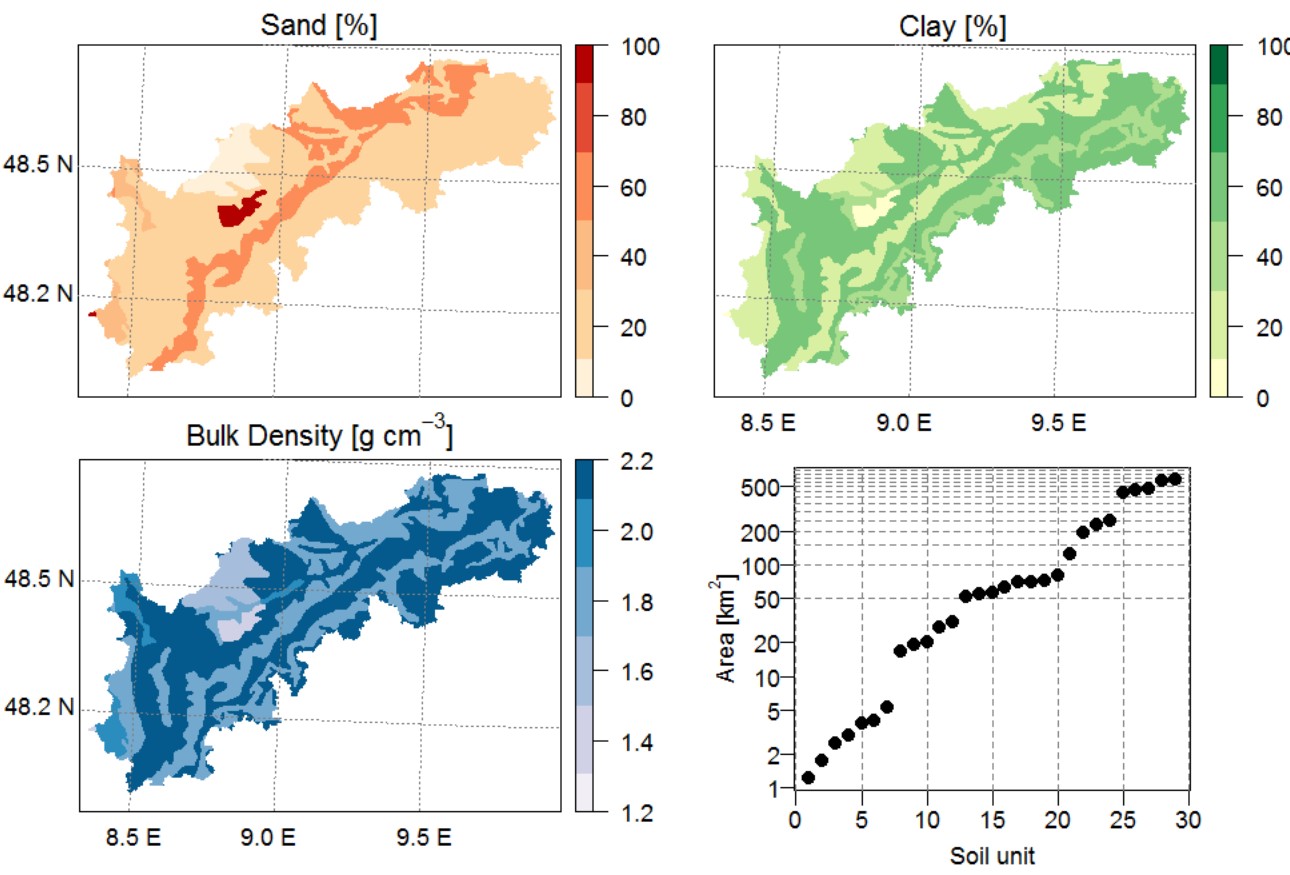

**Figure 3.** Soil maps of sand [%], clay [%] and bulk density [g cm$^{-3}$] and area [km$^2$] of the soil units within the catchment.



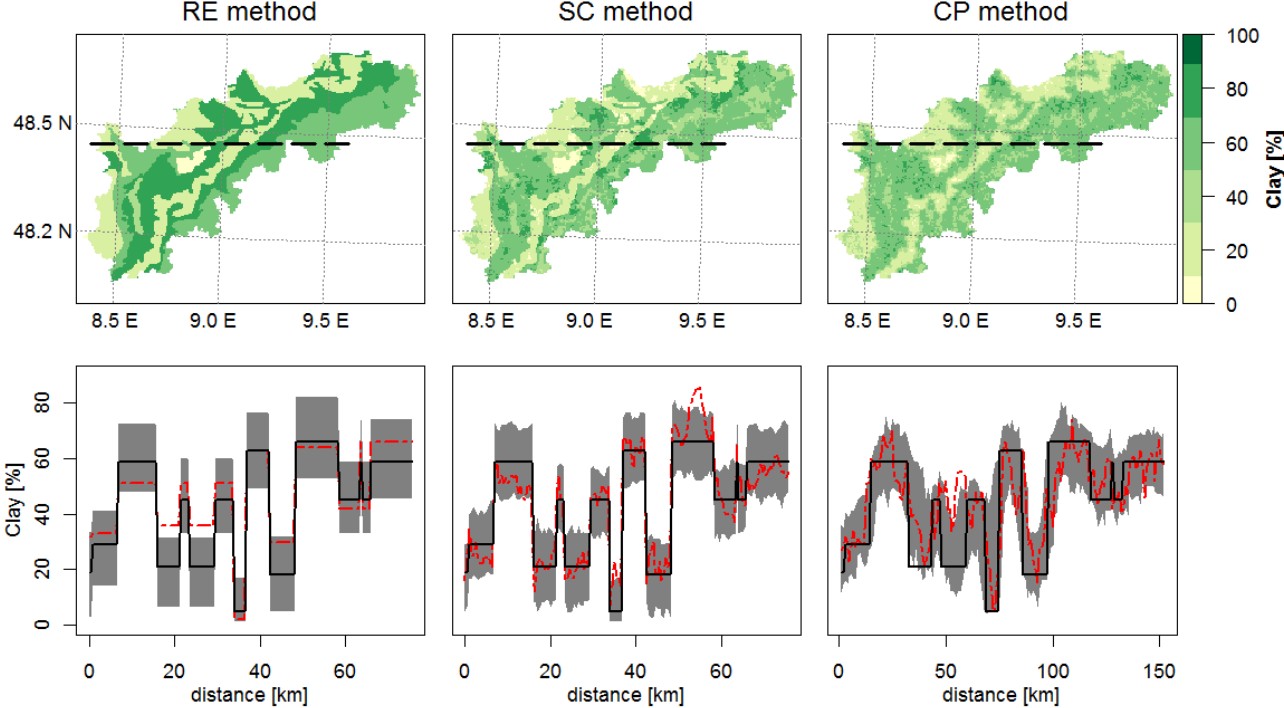

**Figure 4.** Soil realizations obtained for the percentage of clay based on the Random Error method (RE, left column), Spatially Correlated method (SC, middle) and Conditional Points method (CP, right column). The top row shows one realization for each method and the transect (dashed black line). The bottom row depicts the spread of the 100 realizations by using the 5th and 95th percentile for the selected transect (gray area). The red line depicts one realization, whereas the black line shows the percentage of clay by the original soil map.





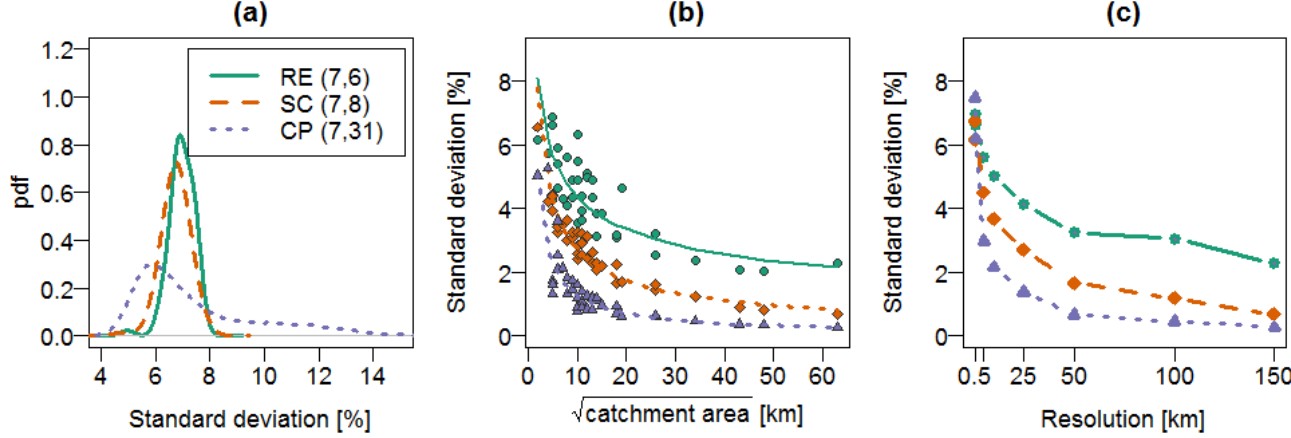

**Figure 5.** (a) Probability distribution of the standard deviation of the clay percentage based on 100 realizations of the soil map calculated for all grid cells and each method (RE = Random Error method; SC = Spatially Correlated method; CP = Conditional Points method). The mean and coefficient of variation of the distribution are indicated in parenthesis. (b) Standard deviation calculated by aggregating the clay percentage at subcatchments with different size. (c) Standard deviation calculated by aggregating the clay percentage at different grid resolutions.





**Figure 6.** Spatial variability of the uncertainty (*CV*) in the simulated model outputs (*Q* = generated runoff; *AET* = evapotranspiration; *SM* = soil moisture; *GWR* = groundwater recharge). On the left column, the results obtained based on the Random Error method (RE) over the entire catchment are depicted together with the position of the transect (dashed black line) and the two grid cells (blue points). On the right column, the *CVs* along the transect within the catchment based on the three perturbation methods (Random Error RE; Spatially Correlated SC; Conditional Points CP) are plotted. Vertical dashed gray lines indicate the position of the grid cells A and B within the transect. Please note that all the plots have individual limits for the y-axis.



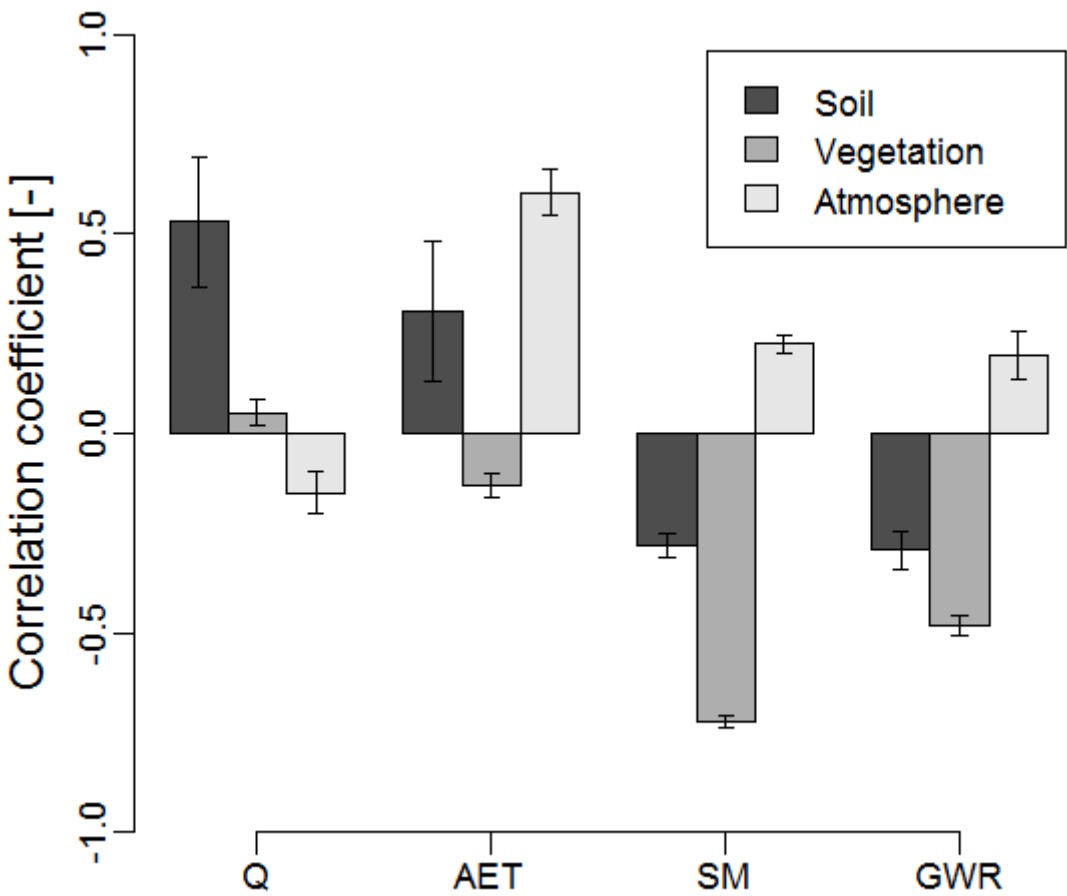

**Figure 7.** Correlation coefficient calculated between the spatial distributions of the uncertainty (*CV*) of the simulated model outputs (*Q* = runoff; *AET* = actual evapotranspiration; *SM* = soil moisture; *GWR* = groundwater recharge) and local environmental conditions (the clay [%] is used to represent the soil; annual mean leaf area index *LAI* [m$^2$ m$^{-2}$] is used to represent the vegetation; cumulative potential evapotranspiration *PET* [mm a$^{-1}$] is used to represent the atmospheric water demand). The bars represent the mean of the correlation coefficients obtained with the three perturbation methods and the error bars the standard deviation.





**Figure 8.** Daily temporal variability of the uncertainty in state and fluxes (*Q* = runoff, *AET* = evapotranspiration, *SM* = soil moisture, *GWR* = groundwater recharge) obtained in two grid cells within the catchment obtained based on the random error method (RE). The mean (black) and the 95% confidence interval (gray) of the ensemble is depicted together with the coefficient of variation (*CV*) calculated at daily time step (red). Note the log y-axis for *Q* and *GWR*. Location of grid cell A and B is shown in Figure 2 and Figure 6.




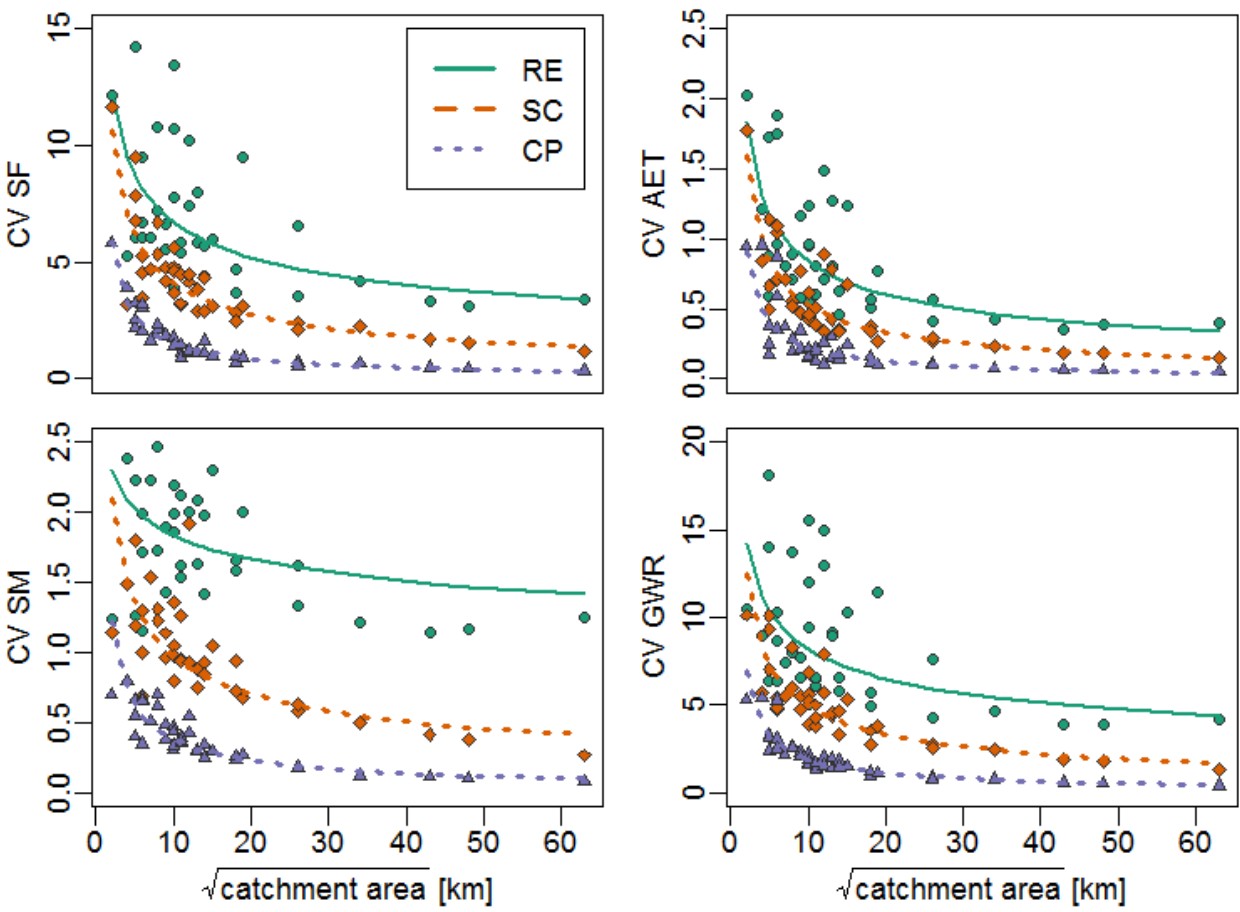

**Figure 9.** Uncertainty, i.e., coefficient of variation (*CV*), of hydrological state and fluxes at catchments with different sizes (*SF* = streamflow, *AET* = evapotranspiration, *SM* = soil moisture, *GWR* = groundwater recharge). Exponential curves are fitted to the data. Please not that all figures have individual limits for the y-axis.





**Figure 10.** Spatio-temporal uncertainty analysis by aggregating the model results at different spatial and temporal resolutions. The three columns refer to the results obtained by (left) Random Error Method - RE, (middle) Spatially Correlated method - SC and (right) Conditional Points method - CP. The rows refers to the different model outputs (i.e., $Q$ = runoff, $AET$ = actual evapotranspiration, $SM$ = soil moisture; $GWR$ = groundwater recharge). Note that a smooth approximation is depicted to facilitate the visualization of the actual $CVs$ values.





**Table 1.** Parameter settings for each perturbation method (Random Error, Spatially Correlated and Conditional Points). Variogram models used for the Spatially Correlated and Conditional Points methods are showed in the supplementary material (Figure S2 and Figure S3, respectively).

| Perturbation method | Parameters | Specific settings |
|---|---|---|
| Random Error | Variance | 50 [$\%^2$] and 0.05 [$g^2\ cm^{-6}$] for texture and bulk density, respectively |
| Spatially correlated | Variograms and co-variograms models | Exponential models (see supplementary material, Figure S2) |
| | Effective variance | 50 [$\%^2$] and 0.05 [$g^2\ cm^{-6}$] for texture and bulk density, respectively |
| | Correlation length | 3 km |
| Conditional Points | Density of samples | 1 sample every 3 x 3 $km^2$ |
| | Variograms and co-variograms models | Two nested exponential models fitted to the empirical variograms and co-variograms (see supplementary material, Figure S3) |





**Table 2.** Overview of the uncertainty analysis presented and discussed.

| | | Perturbation methods | | |
|---|---|---|---|---|
| n. | Uncertainty analysis | 1. Random Error | 2. Spatially Correlated | 3. Conditional Points |
| 1. | Local uncertainty: long term temporal mean of $CV$ at every grid point<br><br>$\mu_{i,t}^m = \frac{1}{N_{ens}}\sum_{j=1}^{N_{ens}} v_{i,t}^{m,j}$<br><br>$\sigma_{i,t}^m = \sqrt{\frac{1}{N_{ens}}\sum_{j=1}^{N_{ens}}\left(v_{i,t}^{m,j} - \mu_{i,t}^m\right)^2}$<br><br>$CV_{i,t}^m = \frac{\sigma_{i,t}^m}{\mu_{i,t}^m}$<br><br>$\overline{CV}_i^m = \frac{1}{T}\sum_{t=1}^{T} CV_{i,t}^m$ | Section 3.2 Figure 6 (left & right black line) | Section 3.2 Figure 6 (right, red line) | Section 3.2 Figure 6 (right, green line) |
| 2. | Local uncertainty: $CV$ at every grid point<br><br>$CV_{i,t}^m$ | Section 3.3 Figure 8 | Section 3.3 | Section 3.3 |
| 3. | Uncertainty by aggregating model output at catchment of different sizes<br><br>$\bar{v}_{sc,t}^{m,j} = \frac{1}{N_{sc}}\sum_{i=1}^{N_{sc}} v_{i,t}^{m,j}$<br><br>$\sigma_{sc,t}^m = \sqrt{\frac{1}{N_{ens}}\sum_{j=1}^{N_{ens}}\left(v_{sc,t}^{m,j} - \mu_{sc,t}^m\right)^2}$<br><br>$CV_{sc,t}^m = \frac{\sigma_{sc,t}^m}{\mu_{sc,t}^m}$ | Section 3.4 Figure 9 (black line) | Section 3.4 Figure 9 (red line) | Section 3.4 Figure 9 (green line) |
| 4. | Uncertainty by aggregating model output at different spatial ($r_d$) and temporal ($t_d$) resolutions<br><br>$\overline{CV^{m,r_d,t_d}} = \frac{1}{T}\sum_{t=1}^{T}\frac{1}{N_r}\sum_{i=1}^{N_r} CV_{i,t}^{m,r_d,t_d}$ | Section 3.5 Figure 10 (left) | Section 3.5 Figure 10 (middle) | Section 3.5 Figure 10 (right) |





**Table 3.** Correlation matrix of the uncertainty (*CV*) of the model outputs (*Q* = generated runoff; A*ET* = actual evapotranspiration; *SM* = soil moisture; *GWR* = groundwater recharge) obtained with the three perturbation methods (Random Error, Spatially Correlated and Conditional Points).

|  |  | *Q* | *AET* | *SM* |
|---|---|---|---|---|
| *AET* | Random Error | 0.3 |  |  |
|  | Spatially Correlated | 0.4 |  |  |
|  | Conditional Points | 0.3 |  |  |
| *SM* | Random Error | -0.1 | 0.0 |  |
|  | Spatially Correlated | -0.1 | -0.1 |  |
|  | Conditional Points | -0.0 | 0.1 |  |
| *GWR* | Random Error | 0.2 | 0.2 | 0.7 |
|  | Spatially Correlated | 0.2 | 0.1 | 0.7 |
|  | Conditional Points | 0.3 | 0.3 | 0.7 |