# Peer review of "Effects of uncertainty in soil properties on simulated hydrological state and fluxes at different spatio-temporal scales"

_Hydrology and Earth System Sciences, 2016_

## Referee Comment (RC1) · Anonymous Referee #1 · 15 Dec 2016

Review of Baroni et al. 2016, STOTEN

General comments

The study analyses uncertainty in soil properties (sand%, clay% and bulk density) using 3 different perturbation methods. The perturbed soil properties are run through the hydrological model mHM in order to evaluate the effect of uncertainty on the soil properties on the simulated hydrological states and fluxes. The uncertainty on the simulated model outputs are afterwards analysed at different spatial and temporal scales. This is an interesting paper and a novel contribution. The paper is technical strong, written in good English and has a good structure. I recommend publishing the paper after minor revision

[Figure]

Specific comments

Novelty: I think you should state more clearly that your study is a novel contribution in respect to both the ways of introducing uncertainty on soil properties (if I understand correctly, this is done more simple in other studies?) and that you take the temporal resolution into account in your analysis (which is not considered in Refsgaard et al. (2016), Hansen et al. (2014), He et al. (2015))

Title: I suggest changing the title to "Effects of uncertainty in soil properties on simulated hydrological state and fluxes at different spatio-temporal scales"

Figure 1 and Page 3, line 22: When first seeing figure 1 and reading the text, I was a bit confused about the transect depicted in the figure. After reading the rest of the article I now understand that it is a horizontal transect through a catchment and not a vertical transect (showing how the sand% change with depth). Could you maybe make this more clear in the text and also in figure1?

Page 3, line 4: I suggest adding some extra text to this sentence, which tells the reader that you are using more sophisticated methods to describe the uncertainty, compared to the studies you mention on page 2 that use more simple assumptions. In this way you clearly indicate that your work is novel.

Page 5, line 11: I do not understand what you mean by "the vertical soil horizons are aggregated to the total soil depth of 2 m"?

Page 5, line 12: How do you define the 29 soil units? Could you show the units on the maps of figure 3?

Page 6, line 21: How is the upscaling done? Is it just taking an area-weighted average of the parameters?

Page 8, line 8 + Figure 2: So are the gauging stations shown on figure 2 "artificial stations" you put in to define the subcatchments you use in analysis #3? If so, could you call them something ells on figure 2 that indicates that these are not real gauging

stations with actual measurements

Page 10, line 32 + page 11, line 1: Are these average CV values across the catchment (15% for Q, 11% for GWR, 3% for SM and 1% for AET) for all the perturbation methods all together (that is how I read the first part of the text) or for the RE method only i.e. the results in figure 6 left (this is how I understand the parenthesis on line 1, p.11)? Please make this more clear in the text.

Page 11, line 14-15 + figure 7: So you calculated correlations coefficients for each of the 3 perturbation methods and then afterwards the average and standard deviation of these R2 (which is plotted on figure 7)? Please specify this in the text and in the figure text.

Page 11, line 20: It looks to me as the pattern in soil moisture uncertainty is very similar to the patterns in clay%? When I visual compare the CV SM map in figure 6 (left) and clay% maps in figure 4.

Page 11, last section: When reading this I was wondering why the AET is not correlation to soil moisture. But you give the explanation on page 12 line 17-18, that AET is close to PET most of the time, and I guess that is why they are not correlated? Maybe you could also mention this explanation on page 11?

Page 12 line 29 + Page 16 line 6: I do not understand what you mean by threshold behaviour/condition?

Page 13, line 18-26 + point 5 in conclusions: You conclude that stream flow, which is an integrated flux, is only sensitive to large spatial structures, whereas the local states and fluxes (i.e. soil moisture, AET, GWR) are sensitive to small scale variations. This makes sense to me. But I would like some more explanation (on page 13) on how you see this from the graphs in figure 9, since that is not clear to me.

Page 14, line 7-26: I found this sections difficult to understand, please consider to rephrase so it is easier to read. Since you are talking about "representative scale" in

the section, I suggest that you present the RES concept already here (you only mention it in the conclusion).

Page 16, line 24-25: I think you should make it more clear, that you have done something new compared to the other studies using the RES approach. I suggest starting the sentence with something like "This study proposes two extensions to the RES approach..."

Technical corrections

Page 1, line 12: Delete "the" in front of "uncertainties"

Page 1, line 12: Change "The methods are applied at the soil map..." to "The methods are applied on the soil map..."

Page 1, line 21: Change to "...(or is not)..."

Page 1, line 24: Delete "the" in front of "uncertainties" and add s on "soil map"

Page 1, line 13: Change "...propagated based on..." to "...propagated through..."

Page 3, line2: Please add "soil" in front of "map" and change "map" to "maps"

Page 3, line 4: Change to "In the present study, we investigate impacts of uncertainty of soil properties on hydrological states and fluxes"

Page 3, line 4-5: change to "Uncertainty in soil properties is..."

Page 3, line 5: Add comma before but

Page 3, line 9-10: Change to "The extent of the impact is expected to decrease with increasing the aggregation area and to disappear at a specific domain size."

Page 3-4, line 31/1: change to "...smaller soil units that have not been detected..."

Page 4, line 27: Change "..can be also.." to ".. can also be.."

Page 5, line 1: Change "field" to "fields"

Page 5, line 12: Please rephrase "...reveals a soil prevalently clay loam..."

Page 5, line 32: Change "...i.e., area smaller than..." to "...i.e., patterns smaller than..."

Page 6, line 11: Change "...and its packages" to "...using add-on packages". Maybe you should write which packages you use?

Page 6, line 23: Please rephrase the sentence. I suggest to change it to "...into 3 layers; the first layer is 5 cm, the second layer is 20 cm and the third has a variable thickness."

Page 6, line 30: I suggest changing ", which covers around 16430 grid cells" to "resulting in 16432 grid cells"

Page 7, line 7: Delete "in" after "yield"

Page 7, line 25: Delete "the" in front of "analysis #1"

Page 8, line 3: Delete "the" in front of "analysis #2"

Page 8, line 5: Add a reference to figure 2 where the location of the grid point are seen

Page 8, line 7-8: Change "In particular, for the analysis #3" to "For use in analysis #3"

Page 8, line 16: Change "cell" to "cells"

Page 8, line 18: Delete "the" in front of "analysis #4"

Page 8, line 18: Change "showed" to "shown"

Page 9, line 9: Change "down row" to "bottom row"

Page 9, line 17: Change to "...highly identifiable and the sharp changes between the units are still preserved."

Page 9, line 22-23: Please rephrase sentence (starting with however), it is difficult to understand.

Page 10, line 5: Change "detailed" to "described"

Page 10, line 8: Add a comma in after magnitude

Page 10, line 10: Change to (i.e., standard deviation > 0 for the resolution of 60 x 60 km2). Maybe the same sentence in line 13 can be shortened?

Page 10, line 11: Delete "the" in front of "spatial scale".

Page 10, line 14: Add a comma in after domain

Page 10, line 29: Change to "...are shown for the transect.."

Page 10, line 32: Change "...over the catchment..." to "...across the catchment..."

Page 11, line 5: Change "affected on" to "affected in"

Page 12, line 23: Change "the first grid cell" to "grid cell A" and "the second grid cell" to "grid cell B"

Page 12, line 25: Delete "the" in front of "grid cell"

Page 14, line 11: Please rephrase "it is notable a certain spread.."

Page 14, line 12: Add s on "catchment"

Page 14, line 23: Change "with reducing the" to "with decreasing"

Page 14, line 25: Change "increasing" to "increasingly"

Page 15, line 3: Change "...across the all number of grid cells..." to "...across all the grid cells..."

Page 15, line 16: Change "enphasises" to "emphasises"

Page 15, line 18: Add d on compensate

Page 15, line 21: Add s on subcatchment
Page 15, line 25: Please add u in "groundwater"

Page 15, line 29: Delete "the" before "soil properties"

Page 15, line 31: Put a " : " after "follow"

Page 16, line 1: Delete "the" in front of uncertainty

Page 16, line 15: Change "different" to "other" (end of line)

Page 16, line 18: Delete "the" in front of "spatial and temporal resolution"

Page 16, line 20: Change "This resolution is referred as…" to "This resolution is referred to as the…"

Page 16, line 33: Please rephrase "…with physical sound…"

Page 17, line 1: Change "soil map" to " a soil map"

Page 17, line 7: Please change last part of line to "… are shown not to be.."

Page 17, line 9: Please add "on stream flow" after "model performance"

Page 17, line 12: Change "input factor" to "input parameters"

Page 17, line 13: Change "soil map" to " a soil map"

Page 17, line 17: change to "…(or is not)…"

––––––––––––––––––––––––––

---

## Author Comment (AC1) · 20 Dec 2016

We thank Reviewer #1 for the very positive feedback and for the constructive and detailed suggestions. We provide point-by-point response to all the specific comments as supplement. Please note that all the technical comments provided by the Reviewer will be integrated in the new version of the manuscript and are not reported in the response document. Thank you and best regards, the Authors

Please also note the supplement to this comment:
http://www.hydrol-earth-syst-sci-discuss.net/hess-2016-657/hess-2016-657-AC1-supplement.pdf

[Figure]

[Figure]

**Supplement:**

**Authors' Response to Reviewer 1**

Authors' response (A, black) to the specific comments of the Reviewer (R, blue).

Specific comments

R: Novelty: I think you should state more clearly that your study is a novel contribution in respect to both the ways of introducing uncertainty on soil properties (if I understand correctly, this is done more simple in other studies?) and that you take the temporal resolution into account in your analysis (which is not considered in Refsgaard et al. (2016), Hansen et al. (2014), He et al. (2015))

A: We thank the Reviewer for underling the novelty of the study. We agree that both aspects, i.e., characterization of the uncertainty in soil properties and temporal resolution, represent a novel contribution and further improvements of current methodologies. For these reasons, we will better emphasize these novelties in the new version of the manuscript. However, we want also to clarify here that there are other perturbation methods that can be considered even more sophisticated (see discussion at Page 4, Line 16-28). But exactly for this reason, these methods are more difficult to be directly applied in e.g., hydrological modelling studies and more simple approaches are usually adopted. By that, we think that the approaches presented in the present studies should be not considered as more complex, but rather as a way to fill a gap among the available methods towards a flexible representation of different uncertainties in soil properties while maintaining a relatively easy integration in hydrological modelling studies. These considerations will be also added in the new version of the manuscript.

R: Title: I suggest changing the title to "Effects of uncertainty in soil properties on simulated hydrological state and fluxes at different spatio-temporal scales"

A: Thank you for the suggestion, we will change the title accordingly.

R: Figure 1 and Page 3, line 22: When first seeing figure 1 and reading the text, I was a bit confused about the transect depicted in the figure. After reading the rest of the article I now understand that it is a horizontal transect through a catchment and not a vertical transect (showing how the sand% change with depth). Could you maybe make this more clear in the text and also in figure1?

A: We will clarify that the depicted line is a horizontal transect to avoid misunderstanding in the new version of the manuscript.

R: Page 3, line 4: I suggest adding some extra text to this sentence, which tells the reader that you are using more sophisticated methods to describe the uncertainty, compared to the studies you mention on page 2 that use more simple assumptions. In this way you clearly indicate that your work is novel.

A: We thank the Reviewer for emphasizing the novelty of the method. As reported in the comment above, we will better clarify in the new version of the manuscript the new aspects of the methods in comparison to the approaches available in literature.

R: Page 5, line 11: I do not understand what you mean by "the vertical soil horizons are aggregated to the total soil depth of 2 m"?

A: The original soil map provides the information for different soil layers to the depth of 2 m. For the present study, this vertical distribution is not accounted for and the vertical average soil property is calculated for each soil unit. We will rephrase the text to avoid misunderstanding.

R: Page 5, line 12: How do you define the 29 soil units? Could you show the units on the maps of figure 3?

A: We use the term soil units to indicate the presence of 29 polygons within the catchment. The term is used to avoid misunderstanding with the term soil class e.g., two soil units could be belonging to the same soil class (e.g., clay loam). But if the Reviewer suggests another terminology, we are open for further suggestions.

R: Page 6, line 21: How is the upscaling done? Is it just taking an area-weighted average of the parameters?

A: The upscaling rules are different for each parameter. The rules were selected based on different specific studies and they are reported and described in Kumar et al. (2013) and Samaniego et al. (2010). However, it has to be noted that in the present study we are actually not upscaling the soil parameters because the generated soil maps are at the same resolution as the model grid i.e., 500 x 500 m$^2$. For this reason, the upscaling rule does not affect the result of the present study. In the new version of the manuscript we will clarify these aspects.

R: Page 8, line 8 + Figure 2: So are the gauging stations shown on figure 2 "artificial stations" you put in to define the subcatchments you use in analysis #3? If so, could you call them something ells on figure 2 that indicates that these are not real gauging stations with actual measurements

A: The locations represent real gauging stations. We preferred to use these positions instead of arbitrary choices. But the Reviewer is right in saying that we did not use the actual streamflow measurements for comparison but we rather use the locations only to define the drainage areas. We will clarify this in the new version of the manuscript.

R: Page 10, line 32 + page 11, line 1: Are these average CV values across the catchment (15% for Q, 11% for GWR, 3% for SM and 1% for AET) for all the perturbation methods all together (that is how I read the first part of the text) or for the RE method only i.e. the results in figure 6 left (this is how I understand the parenthesis on line 1, p.11)? Please make this more clear in the text.

A: Yes, they are and additional information will be added in the new version of the manuscript to avoid misunderstanding.

Page 11, line 14-15 + figure 7: So you calculated correlations coefficients for each of the 3 perturbation methods and then afterwards the average and standard deviation of these R2 (which is plotted on figure 7)? Please specify this in the text and in the figure text.

A: Yes, we did. We will better specify this information in text and legend.

Page 11, line 20: It looks to me as the pattern in soil moisture uncertainty is very similar to the patterns in clay%? When I visual compare the CV SM map in figure 6 (left) and clay% maps in figure 4.

A: We checked again the correlation coefficients calculated between CVs and clay. The values are correct. From visual comparison, we see, on the one hand, that one soil unit is remarkable visible with high values in the CV SM map, i.e., one long soil unit crossing the entire catchment from south-west to north-east. On the other hand, however, other locations are not highly correlated. For these reasons, we believe that the visual comparison could be misleading for the average correlation over the entire catchment as it is quantify instead by the calculated correlation coefficient.

R: Page 11, last section: When reading this I was wondering why the AET is not correlation to soil moisture. But you give the explanation on page 12 line 17-18, that AET is close to PET most of the time, and I guess that is why they are not correlated? Maybe you could also mention this explanation on page 11?

A: In this section we discuss the correlation and we still do not provide the actual values of SM and AET to justify the behavior. For this reason, we rather prefer to stick to the results presented i.e., to discuss the correlations here and to leave the interpretation for later in the text where the actual values are depicted. However, we will refer at page 12 to page 11 for giving reasoning for the observed correlations.

R: Page 12 line 29 + Page 16 line 6: I do not understand what you mean by threshold behaviour/condition?

A: The relation between soil moisture and fluxes is non-linear and based on threshold conditions. For instance, the fast interflow runoff component is activated if a certain soil water content is available. For this reason, if we have uncertain SM but these values are below this soil water content, we will not have uncertainty on simulated runoff. It can now happen to have the same uncertainty in SM but in a range where SM is above this water content. In this case, runoff is also affected. A similar consideration holds for the relation between SM and AET but for soil drying conditions i.e., in case SM is below a certain soil water content, AET is affected. For these reasons, the uncertainty in SM does not always reflect the uncertainty in the other fluxes. We will extend this discussion in the text to better clarify the hydrological behavior. It has also to be noted that these considerations are not specific to the hydrological model used in the present study but they hold for most of the hydrological models.

R: Page 13, line 18-26 + point 5 in conclusions: You conclude that stream flow, which is an integrated flux, is only sensitive to large spatial structures, whereas the local states and fluxes (i.e. soil moisture, AET, GWR) are sensitive to small scale variations. This makes sense to me. But I would like some more explanation (on page 13) on how you see this from the graphs in figure 9, since that is not clear to me.

A: We agree with the Reviewer that we did not clarify enough how it is possible to infer these conclusions as they are actually given by comparing both Figure 5 and Figure 9. Figure 5 presents the uncertainty introduced in the soil properties by each method. Here we show how RE method perturb long spatial structure while CP method only small scale features. Figure 9 represent the uncertainty in the model output. In this case, we can look at the streamflow of the catchment (e.g., SF CV for catchment > 60 x 60 km$^2$) and see that this model output is strongly perturbed by the RE method, and for that, only by the long spatial structures. In the same figure we can look at the uncertainty in SM or AET at the model resolution or, eventually, as they could be measured in the field e.g., catchment < 1 x 1 km$^2$. These model outputs are affected also by the uncertainty introduced by the CP method. For this reason, these local state and fluxes are sensitive to small scale variations.

This explanation will be extended and integrated in the new version of the manuscript. To this end, we rather realized that it is more convenient to first discuss the characteristic correlation lengths introduced by each perturbation method (Page 13 line 30 – Page 14 line 26) and later to extend the discussion to the implications for the specific model applications (Page 13, line 18-26). For this reason we will reorganize this section.

Page 14, line 7-26: I found this sections difficult to understand, please consider to rephrase so it is easier to read. Since you are talking about "representative scale" in the section, I suggest that you present the RES concept already here (you only mention it in the conclusion).

A: We agree with the Reviewer that the discussion requires further extension. We tried to combine relevant results obtained in different disciplines (i.e., the REA scale, the RES concept, the ergodicity concept) but we rather provide only limited information about that while referring to additional references for further details. In the new version of the manuscript we will elaborate the discussion to better clarify the different concepts also within this manuscript.

R: Page 16, line 24-25: I think you should make it more clear, that you have done something new compared to the other studies using the RES approach. I suggest starting the sentence with something like "This study proposes two extensions to the RES approach…"

A: Yes, we will emphasize more the novelty of the present study. Thanks.

Technical comments (not listed)

A: Please note that all the technical comments provided by the Reviewer will be integrated in the new version of the manuscript and are not reported in this response document.

---

## Referee Comment (RC2) · Anonymous Referee #2 · 13 Jan 2017

The contribution analyses the uncertainty of model fluxes with regards the soil properties. The setup is as following: Different distributed soil properties – the percentage of sand and clay, and bulk density – are linked through a functional relationship with the parameters of the hydrological model mHM. Changing the soil properties has thus a direct impact on the model states and fluxes. The variability of the fluxes with respect to changes in the soil properties is investigated by perturbing the different properties with three different methods. The resulting ensemble of model states and fluxes is then analysed with regard to the uncertainties at different spatial and temporal scales. The contribution is novel, well written and logically structured. It is well suited for a publication in HESS. As is so often the case some small adaptions could be made. I therefore

recommend publishing it after some minor revisions.

—

Specific comments:

Parameter maps. Currently the mentioned link between soil properties, model parameters and model fluxes is never explicitly stated within the paper. The reader is left a alone with a set of references (see: p. 6, l. 25f) that lead to a detailed description of the model with all its pedotransfer functions. I believe that many readers would appreciate if the (most) relevant pedotransfer functions are explicitly mentioned (e.g. within a table). Following that reasoning, one would also like to see some maps (they should be readily available) of the change of parameters with respect to a change in soil properties. Such maps could be made in a similar fashion as Fig. 4, where a perturbation example for percentage of clay is mapped. This would make it easier for readers to connect the information shown in Fig. 3, Fig. 4 and Fig. 7. The paper does already provide ample amounts of supplementary material, so it could at least be added there.

P 5. l. 24. This comment might be a little nit-picky. It would be nice if the authors would elaborate a little about the mentioned bounds. One or two sentences would already suffice to make the picture clearer: Besides the 100% upper bound in the sum, the soil properties are (obviously) also bound at 0%. Furthermore there sum always results in 100%. That is (of course) the reason why the silt does not need to be perturbed directly, as it is fully defined by the other two textural classes. Now, this is all clear to readers but I think it would be good to be mentioned explicitly. Furthermore, one might expect that these bounds mess with Gaussian noise in some minor way. Intuitively one would expect that this lowers the uncertainty in some areas of the basin. This notion is dispelled in Fig. 5. It shows clearly that these theoretic influences are as good as non-existent. They are not are not visible at all! Nevertheless, there are areas in Fig. 3 (and 4) where the soil properties are exactly at these bounds – the sand is at 100% and the clay is at 0%. I think one should at least be clarify these aspects, even if they

seem to be irrelevant for the resulting analysis (as shown in Fig. 6). On the other hand, it could be that I am missing something.

P. 5, l. 21-22 and Table 1. The noise is defined by its variance. I would propose to use the standard deviation instead. This would (a) make it easier for readers to interpret (clearer units, simpler dispersion summary); (b) bring closer to the units used for the analysis (Fig. 5 shows the standard deviation (!) of the clay-ensemble); and (c) sync in a rather abstract way with the uncertainty quantification (the coefficient of variation is defined through the standard deviation).

P. 13, l. 18-22. These sentences need to be reformulated somehow. The intention or setup is not clear to me. In concrete: The phrasing "streamflow at the catchment outlet" is used twice, which makes it difficult to understand.

P. 17, l.5-20. The section is understandable as such, but could be rephrased to make it easier to read. As I understand it, the gist is that fine-grained soil information is important for local states and fluxes but not for integrated ones. As it stands now, one reads at first that the fine soil resolution is not important for model performance (p.17, l. 9), only to read a view later that the fine soil resolution is important for model performance (p.17, l. 14-15). Readers will infer the meaning from the context, but the phrasing seems to be needlessly difficult.

---

## Author Comment (AC2) · 18 Jan 2017

**Authors' Response to Reviewer 2**

We thank the Reviewer for the positive feedback and for the additional suggestions provided to improve the manuscript. We will integrate all of them in the new version of the manuscript as discussed below.

Thank you and best regards,
The Authors

Authors' response (A, black) to the comments of the Reviewer (R, blue).

Specific comments:

R: Parameter maps. Currently the mentioned link between soil properties, model parameters and model fluxes is never explicitly stated within the paper. The reader is left a alone with a set of references (see: p. 6, l. 25f) that lead to a detailed description of the model with all its pedotransfer functions. I believe that many readers would appreciate if the (most) relevant pedotransfer functions are explicitly mentioned (e.g. within a table). Following that reasoning, one would also like to see some maps (they should be readily available) of the change of parameters with respect to a change in soil properties. Such maps could be made in a similar fashion as Fig. 4, where a perturbation example for percentage of clay is mapped. This would make it easier for readers to connect the information shown in Fig. 3, Fig. 4 and Fig. 7. The paper does already provide ample amounts of supplementary material, so it could at least be added there.

A: we agree with the Reviewer that additional information regarding the pedotransfer functions (PTFs) will help the readers to better connect the perturbation of the soil properties (texture and bulk density) to the simulated state and fluxes. For this reason, (1) we will provide a list of most relevant pedotransfer functions (PTFs) integrated in the model, and (2) we will provide some maps of the soil hydraulic parameters estimated based on these PTFs in the supplementary material.

R: P 5. l. 24. This comment might be a little nit-picky. It would be nice if the authors would elaborate a little about the mentioned bounds. One or two sentences would already suffice to make the picture clearer: Besides the 100% upper bound in the sum, the soil properties are (obviously) also bound at 0%. Furthermore there sum always results in 100%. That is (of course) the reason why the silt does not need to be perturbed directly, as it is fully defined by the other two textural classes. Now, this is all clear to readers but I think it would be good to be mentioned explicitly. Furthermore, one might expect that these bounds mess with Gaussian noise in some minor way. Intuitively one would expect that this lowers the uncertainty in some areas of the basin. This notion is dispelled in Fig. 5. It shows clearly that these theoretic influences are as good as non-existent. They are not are not visible at all! Nevertheless, there are areas in Fig. 3 (and 4) where the soil properties are exactly at these bounds – the sand is at 100% and the clay is at 0%. I think one should at least be clarify these aspects, even if they seem to be irrelevant for the resulting analysis (as shown in Fig. 6). On the other hand, it could be that I am missing something.

A: we agree that forcing the texture perturbation to bounds (i.e., 0 < texture < 100) (i) could mess with Gaussian noise and (ii) could lower the uncertainty in areas of the basin where the actual values are close to the bounds. As also pointed by the Reviewer, these characteristics are not relevant in our study. This is because the noise used (variance) is relatively low and the areas within the catchment where the actual values are close to these bounds (0 or 100%) are relatively

limited. Still, these characteristics of the perturbation methods could be relevant for other studies i.e., when the noise introduced is higher or extreme texture values are more frequently presented within the catchment. For these reasons, we think that the comment of the Reviewer is relevant for a more general understanding of the perturbation methods and we will add this information in the revised manuscript.

R: P. 5, l. 21-22 and Table 1. The noise is defined by its variance. I would propose to use the standard deviation instead. This would (a) make it easier for readers to interpret (clearer units, simpler dispersion summary); (b) bring closer to the units used for the analysis (Fig. 5 shows the standard deviation (!) of the clay-ensemble); and (c) sync in a rather abstract way with the uncertainty quantification (the coefficient of variation is defined through the standard deviation).

A: we agree that the use of the standard deviation will simplify the interpretation and it will be consistent with Figure 5. For this reason we will modify these values in the revised version of the manuscript. It has to be noted, however, that the variograms (e.g., exponential model used for the spatial correlated method) are usually defined by variance and correlation length. To our knowledge, it would be rather uncommon to present the variogram model in terms of standard deviation. And for this reason, the variograms depicted in the supplementary material (Figure S2 and Figure s3) will be not modified.

P. 13, l. 18-22. These sentences need to be reformulated somehow. The intention or setup is not clear to me. In concrete: The phrasing "streamflow at the catchment outlet" is used twice, which makes it difficult to understand.

A: streamflow at the catchment outlet is sensitive to the perturbation of long spatial structure while it is not sensitive to small scale soil perturbation. This result is shown in Figure 9 comparing the CV of SF obtained for the bigger catchment (64x64 $km^2$) based on the RE and CP methods, respectively. A similar lack of clarity in the discussion of these results was underlined also by the other Reviewer. Overall, we realized that it is more convenient for a better understanding to first discuss the characteristic correlation lengths introduced by each perturbation method (Page 13 line 30 – Page 14 line 26) and later to extend the discussion to the implications for the specific model applications (Page 13, line 18-26). We will reorganize this section in the revised manuscript, accordingly.

P. 17, l.5-20. The section is understandable as such, but could be rephrased to make it easier to read. As I understand it, the gist is that fine-grained soil information is important for local states and fluxes but not for integrated ones. As it stands now, one reads at first that the fine soil resolution is not important for model performance (p.17, l. 9), only to read a view later that the fine soil resolution is important for model performance (p.17, l. 14-15). Readers will infer the meaning from the context, but the phrasing seems to be needlessly difficult.

A: the gist is very well summarized by the Reviewer. We will phrase the text accordingly to avoid possible misunderstanding.

---

## Author Response (AR1)

Dear Editor,

as requested, we provide point-by-point response to all the comments, the revised manuscript and supplement material with track changes (attached below) and without (in separate files). In the response letter, Authors' response are identified by AR (black color) while the comments of the Reviewer are identified by RC (blue color). Page (P) and lines (L) refer to the revised manuscript in track-changes to facilitate the reading.

The manuscript has been revised based on all the general and specific comments provided by the two Reviewers. Figures and tables were also updated accordingly. Please note that Figures 4 has been also replaced by correcting the x-axis labels for the CP method.

Thank you and best regards, The Authors

**Authors' Response to Reviewer 1**

**Specific comments**

RC. Novelty: I think you should state more clearly that your study is a novel contribution in respect to both the ways of introducing uncertainty on soil properties (if I understand correctly, this is done more simple in other studies?) and that you take the temporal resolution into account in your analysis (which is not considered in Refsgaard et al. (2016), Hansen et al. (2014), He et al. (2015))

AR. We highlighted the novelty of these aspects in the introduction (P3L7-9) and in the conclusion sections (P18) of the revised manuscript.

RC. Title: I suggest changing the title to "Effects of uncertainty in soil properties on simulated hydrological state and fluxes at different spatio-temporal scales"

AR. The title has been changed.

RC. Figure 1 and Page 3, line 22: When first seeing figure 1 and reading the text, I was a bit confused about the transect depicted in the figure. After reading the rest of the article I now understand that it is a horizontal transect through a catchment and not a vertical transect (showing how the sand% change with depth). Could you maybe make this more clear in the text and also in figure 1?

AR. We specified that is a horizontal transect through a soil map in the text (P3L25) and in the caption of Figure 1. Note that Figure 1 was replaced with a new figure where x-axis of the horizontal transect is expressed in km to better identify that is a horizontal transect and not a vertical soil profile. Names of the methods were also modified to be more consistent with the acronyms used in the manuscript.

RC. Page 3, line 4: I suggest adding some extra text to this sentence, which tells the reader that you are using more sophisticated methods to describe the uncertainty, compared to the studies you mention on page 2 that use more simple assumptions. In this way you clearly indicate that your work is novel.

AR. We added some extra text to better clarify that we are presenting also a new method (P3L7-9).

RC: Page 5, line 11: I do not understand what you mean by "the vertical soil horizons are aggregated to the total soil depth of 2 m"?

AR. The text was rephrased (see P5L16-17).

RC. Page 5, line 12: How do you define the 29 soil units? Could you show the units on the maps of figure 3?

AR. We specified that the term soil units refer to polygons within the catchment (see P5L16-19).

RC. Page 6, line 21: How is the upscaling done? Is it just taking an area-weighted average of the parameters?

AR. We specified in the revised manuscript (P7L1-3) that the upscaling is done based on different average rules for each parameter (e.g. arithmetic, geometric, maximum). The rules are reported and described in details in Kumar et al. (2013) and Samaniego et al. (2010) and we referred to these studies for additional information.

RC. Page 8, line 8 + Figure 2: So are the gauging stations shown on figure 2 "artificial stations" you put in to define the subcatchments you use in analysis #3? If so, could you call them something else on figure 2 that indicates that these are not real gauging stations with actual measurements

AR: The positions represent real gauging stations. We specified in the revised manuscript (P8L22-23) that the positions of the gauging stations were used to define the subcatchments.

RC. Page 10, line 32 + page 11, line 1: Are these average CV values across the catchment (15% for Q, 11% for GWR, 3% for SM and 1% for AET) for all the perturbation methods all together (that is how I read the first part of the text) or for the RE method only i.e. the results in figure 6 left (this is how I understand the parenthesis on line 1, p.11)? Please make this more clear in the text.

AR. The text was rephrased in the revised manuscript (see P11L17-19) to clarify that average CVs estimated across the catchment are the same for each perturbation method. Differences are detected only in transition between the soil units.

RC. Page 11, line 14-15 + figure 7: So you calculated correlations coefficients for each of the 3 perturbation methods and then afterwards the average and standard deviation of these R2 (which is plotted on figure 7)? Please specify this in the text and in the figure text.

AR. We added additional text in the revised manuscript (P12L1-2) to clarify that the average and standard deviation of the 3 perturbation methods are plotted. In the capture legend of Figure 7 it was already indicated that the bars represent the mean of the correlation coefficients obtained with the three perturbation methods and the error bars the standard deviation. For this reason no changes were done there.

RC. Page 11, line 20: It looks to me as the pattern in soil moisture uncertainty is very similar to the patterns in clay%? When I visual compare the CV SM map in figure 6 (left) and clay% maps in figure 4.

AR. We checked again the correlation coefficients calculated between CVs and clay. The values are correct. From visual comparison, we see, on the one hand, that one soil unit is remarkable visible with high values in the CV SM map, i.e., one long soil unit crossing the entire catchment from south-west to north-east. On the other hand, however, other locations are not highly correlated. For these reasons, we believe that the visual comparison could be misleading for the average correlation over the entire catchment as it is quantify instead by the calculated correlation coefficient. For this reason no changes were done in the revised manuscript.

RC. Page 11, last section: When reading this I was wondering why the AET is not correlation to soil moisture. But you give the explanation on page 12 line 17-18, that AET is close to PET most of the time, and I guess that is why they are not correlated? Maybe you could also mention this explanation on page 11?

AR. In section 3.3 we discuss the correlation while in section 3.4 we provide the actual values of SM and AET. For this reason, we preferred to stick to the results presented i.e., to discuss only the correlations in section 3.3 and to extend the discussion later by reminding the correlations found (see P13L8-9).

RC. Page 12 line 29 + Page 16 line 6: I do not understand what you mean by threshold behaviour/condition?

AC. Additional text was added to clarify the threshold conditions (see P13L20-22).

RC. Page 13, line 18-26 + point 5 in conclusions: You conclude that stream flow, which is an integrated flux, is only sensitive to large spatial structures, whereas the local states and fluxes (i.e. soil moisture, AET, GWR) are sensitive to small scale variations. This makes sense to me. But I would like some more explanation (on page 13) on how you see this from the graphs in figure 9, since that is not clear to me.

AR. This conclusion is supported by comparing the results presented in both Figure 5 and Figure 9. Figure 5 presents the uncertainty introduced in the soil properties by each method. Here we show how RE method perturb long spatial structure while CP method only small scale features. Figure 9 represent the uncertainty in the model output. In this case, we can look at the uncertainty in the simulated streamflow of the entire catchment (e.g., SF CV for catchment > 60 x 60 km2) and see that this model output is strongly perturbed by the RE method, and for that, only by the long spatial structures. In the same figure we can look at the uncertainty in SM or AET at the model resolution or, eventually, as they could be measured in the field e.g., catchment <  $1 \times 1 \text{ km}^2$ . These model outputs are affected also by the uncertainty introduced by the CP method. For this reason, these local states and fluxes are sensitive to small scale variations. This explanation is now extended and better integrated in the revised manuscript. To this end, we rather found more convenient to first discuss the characteristic correlation lengths introduced by each perturbation method in section 3.4 and to move in section 3.5 the discussion about the implications for the specific model applications.

RC. Page 14, line 7-26: I found this section difficult to understand, please consider to rephrase so it is easier to read. Since you are talking about "representative scale" in the section, I suggest that you present the RES concept already here (you only mention it in the conclusion).

AR. The section was revised by introducing the RES concept and by rephrasing the discussion accordingly.

RC: Page 16, line 24-25: I think you should make it more clear, that you have done something new compared to the other studies using the RES approach. I suggest starting the sentence with something like "This study proposes two extensions to the RES approach..."

AR. The conclusion was rephrased to better highlight the novelty of the study (P18L3-20).

**Technical comments (not listed)**

Page 1, line 12: Delete "the" in front of "uncertainties" AC. Done. Page 1, line 12: Change "The methods are applied at the soil map: ::" to "The methods are applied on the soil map: :: " AR. Done. Page 1, line 21: Change to ": : :(or is not): : :" AR. Done. Page 1, line 24: Delete "the" in front of "uncertainties" and add s on "soil map" AR. Done. Page 1, line 13: Change ": : : propagated based on: : :" to ": : : propagated through: : :" AR. Done. Page 3, line2: Please add "soil" in front of "map" and change "map" to "maps" AR. Done. Page 3, line 4: Change to "In the present study, we investigate impacts of uncertainty of soil properties on hydrological states and fluxes" AR. Done. Page 3, line 4-5: change to "Uncertainty in soil properties is: ::" AR. Done. Page 3. line 5: Add comma before but AR. Done. Page 3, line 9-10: Change to "The extent of the impact is expected to decrease with increasing the aggregation area and to disappear at a specific domain size." AR. Done. Page 3-4, line 31/1: change to ": : :smaller soil units that have not been detected: : :" AR. Done. Page 4, line 27: Change "..can be also.." to ".. can also be.." AR. Done. Page 5, line 1: Change "field" to "fields" AR. Done. Page 5, line 12: Please rephrase ": : : reveals a soil prevalently clay loam..." AR. Done. Page 5, line 32: Change ": : :i.e., area smaller than: : :" to ": : :i.e., patterns smaller than: : :" AR. Done. Page 6, line 11: Change ": : : and its packages" to ": : : using add-on packages". Maybe you should write which packages you use? AR. Done. We cited the reference of the package i.e., Pebesma, E.J., 2004. Multivariable geostatistics in S: the gstat package. Comput. Geosci. 30, 683–691. doi:10.1016/j.cageo.2004.03.012 Page 6, line 23: Please rephrase the sentence. I suggest to change it to ": : :into 3 layers; the first layer is 5 cm, the second layer is 20 cm and the third has a variable thickness." AR. Changed as suggested.

Page 6, line 30: I suggest changing ", which covers around 16430 grid cells" to "resulting in 16432 grid cells"

Page 7, line 7: Delete "in" after "yield" AR. Done. Page 7, line 25: Delete "the" in front of "analysis #1" AR. Done. Page 8, line 3: Delete "the" in front of "analysis #2" AR. Done. Page 8, line 5: Add a reference to figure 2 where the location of the grid point are seen AR. Done. Page 8, line 7-8: Change "In particular, for the analysis #3" to "For use in analysis #3" AR. Done. Page 8, line 16: Change "cell" to "cells" AR. Done. Page 8, line 18: Delete "the" in front of "analysis #4" AR. Done. Page 8, line 18: Change "showed" to "shown" AR. Done. Page 9, line 9: Change "down row" to "bottom row" AR. Done. Page 9, line 17: Change to ": : :highly identifiable and the sharp changes between the units are still preserved." AR. Done. Page 9, line 22-23: Please rephrase sentence (starting with however), it is difficult to understand. AR. Done. Page 10, line 5: Change "detailed" to "described" AR. Done. Page 10, line 8: Add a comma in after magnitude AR. Done. Page 10, line 10: Change to (i.e., standard deviation > 0 for the resolution of 60 x 60 km2). Maybe the same sentence in line 13 can be shortened? AR. Line 10 changed and line 13 shortened. Page 10, line 11: Delete "the" in front of "spatial scale". AR. Done. Page 10, line 14: Add a comma in after domain AR. Done. Page 10, line 29: Change to ": : : are shown for the transect.." AR. Done. Page 10, line 32: Change ": : :over the catchment..." to ": : :across the catchment..." AR. Done. Page 11, line 5: Change "affected on" to "affected in" AR. Done. Page 12, line 23: Change "the first grid cell" to "grid cell A" and "the second grid cell" to "grid cell B" AR. Done. Page 12, line 25: Delete "the" in front of "grid cell" AR. Done. Page 14, line 11: Please rephrase "it is notable a certain spread.." AR. Done. Page 14, line 12: Add s on "catchment" AR. Done. Page 14, line 23: Change "with reducing the" to "with decreasing" AR. Done.

Page 14, line 25: Change "increasing" to "increasingly" AR. Done. Page 15, line 3: Change ": : : across the all number of grid cells: : : " to ": : : across all the grid cells: : : " AR. Done. Page 15, line 16: Change "enphasises" to "emphasises" AR. Done. Page 15, line 18: Add d on compensate AR. Done. Page 15, line 21: Add s on subcatchment AR. Done. Page 15, line 25: Please add u in "groundwater" AR. Done. Page 15, line 29: Delete "the" before "soil properties" AR. Done. Page 15, line 31: Put a ": " after "follow" AR. Done. Page 16, line 1: Delete "the" in front of uncertainty AR. Done. Page 16, line 15: Change "different" to "other" (end of line) AR. Done. Page 16, line 18: Delete "the" in front of "spatial and temporal resolution" AR. Done. Page 16, line 20: Change "This resolution is referred as: ::" to "This resolution is referred to as the: ::" AR. Done. Page 16, line 33: Please rephrase ": : : with physical sound: : : " AR. Done. Page 17, line 1: Change "soil map" to " a soil map" AR. Done. Page 17, line 7: Please change last part of line to ": : : are shown not to be.." AR. Done. Page 17, line 9: Please add "on stream flow" after "model performance" AR. Done. Page 17, line 12: Change "input factor" to "input parameters" AR. Done. Page 17, line 13: Change "soil map" to " a soil map" AR. Done. Page 17, line 17: change to ": : :(or is not): : :" AR. Done.

**Authors' Response to Reviewer 2**

**Specific comments:**

RC. Parameter maps. Currently the mentioned link between soil properties, model parameters and model fluxes is never explicitly stated within the paper. The reader is left alone with a set of references (see: p. 6, l. 25f) that lead to a detailed description of the model with all its pedotransfer functions. I believe that many readers would appreciate if the (most) relevant pedotransfer functions are explicitly mentioned (e.g. within a table). Following that reasoning, one would also like to see some maps (they should be readily available) of the change of parameters with respect to a change in

soil properties. Such maps could be made in a similar fashion as Fig. 4, where a perturbation example for percentage of clay is mapped. This would make it easier for readers to connect the information shown in Fig. 3, Fig. 4 and Fig. 7. The paper does already provide ample amounts of supplementary material, so it could at least be added there.

AR. We provided in the supplementary material the list of the pedotransfer functions integrated in the model (see Table S1), and two additional maps showing saturated water content and saturated hydraulic conductivity estimated based on these PTFs. Please note that the plots were reorganized to facilitate the comparison (See Figure S4-S5-S6). These new table and figures were also cited in the manuscript (see P7L8 and P9L24-26).

RC. P 5. l. 24. This comment might be a little nit-picky. It would be nice if the authors would elaborate a little about the mentioned bounds. One or two sentences would already suffice to make the picture clearer: Besides the 100% upper bound in the sum, the soil properties are (obviously) also bound at 0%. Furthermore there sum always results in 100%. That is (of course) the reason why the silt does not need to be perturbed directly, as it is fully defined by the other two textural classes. Now, this is all clear to readers but I think it would be good to be mentioned explicitly. Furthermore, one might expect that these bounds mess with Gaussian noise in some minor way. Intuitively one would expect that this lowers the uncertainty in some areas of the basin. This notion is dispelled in Fig. 5. It shows clearly that these theoretic influences are as good as non-existent. They are not are not visible at all! Nevertheless, there are areas in Fig. 3 (and 4) where the soil properties are exactly at these bounds – the sand is at 100% and the clay is at 0%. I think one should at least be clarify these aspects, even if they seem to be irrelevant for the resulting analysis (as shown in Fig. 6). On the other hand, it could be that I am missing something.

AR. We think the comment is relevant for a more general understanding of the perturbation methods and we added this information in the revised manuscript (see P6L17-22).

RC. P. 5, l. 21-22 and Table 1. The noise is defined by its variance. I would propose to use the standard deviation instead. This would (a) make it easier for readers to interpret (clearer units, simpler dispersion summary); (b) bring closer to the units used for the analysis (Fig. 5 shows the standard deviation (!) of the clay-ensemble); and (c) sync in a rather abstract way with the uncertainty quantification (the coefficient of variation is defined through the standard deviation).

AR. We changed the values by referring to the standard deviation in the revised version of the manuscript. Please note, however, that the variograms (e.g., exponential model used for the spatial correlated method) are usually defined by variance and correlation length. To our knowledge, it would be rather uncommon to present the variogram model in terms of standard deviation. And for this reason, the variograms depicted in the supplementary material (Figure S2 and Figure s3) are not modified.

RC. P. 13, l. 18-22. These sentences need to be reformulated somehow. The intention or setup is not clear to me. In concrete: The phrasing "streamflow at the catchment outlet" is used twice, which makes it difficult to understand.

AR. For clarity, in the revised manuscript, we decided to focus the discussion to the characteristic correlation lengths introduced by each perturbation method in section 3.4. In section 3.5 we focused the discussion to the implication for the model application by referring to Figure 10. By that, text was extensively revised.

RC. P. 17, l.5-20. The section is understandable as such, but could be rephrased to make it easier to read. As I understand it, the gist is that fine-grained soil information is important for local states and fluxes but not for integrated ones. As it stands now, one reads at first that the fine soil resolution is not important for model performance (p.17, l. 9), only to read a view later that the fine soil resolution is important for model performance (p.17, l. 14-15). Readers will infer the meaning from the context, but the phrasing seems to be needlessly difficult.

AR. The text was rephrased to make the message easier to read (see P19L1-17).

[revised manuscript text omitted]